# Deep desiccation of soils observed by long-term high-resolution measurements on a large inclined lysimeter

Markus Merk[1], Nadine Goeppert[1], and Nico Goldscheider[1]

[1]Institute of Applied Geosciences (AGW), Karlsruhe Institute of Technology (KIT), Kaiserstr. 12, 76131 Karlsruhe, Germany

**Correspondence:** (markus.merk@kit.edu)

**Abstract.** Availability of long-term and high-resolution measurements of soil moisture is crucial when it comes to understanding all sorts of changes to past soil moisture variations and the prediction of future dynamics. This is particularly true in a world struggling against climate change and its impacts on ecology and economy. Feedback mechanisms between soil moisture dynamics and meteorological influences are key factors when it comes to understanding the occurrence of drought events. We used long-term high-resolution measurements of soil moisture on a large inclined lysimeter at a test site near Karlsruhe, Germany. The measurements indicate (i) a seasonal evaporation depth of over two meters. Statistical analysis and linear regressions indicate (ii) a significant decrease in soil moisture levels over the past two decades. This decrease is most pronounced at the start and the end of the vegetation period. Furthermore, Bayesian change point detection revealed (iii), that this decrease is not uniformly distributed over the complete observation period. Largest changes occur at tipping points during years of extreme drought, with significant changes to the subsequent soil moisture levels. This change affects not only the overall trend in soil moisture, but also the seasonal dynamics. A comparison to modeled data showed (iv) that the occurrence of deep desiccation is not merely dependent on the properties of the soil but is spatially heterogeneous. The study highlights the importance of soil moisture measurements for the understanding of moisture fluxes in the vadose zone.

## 1 Introduction

The understanding of soil moisture dynamics and its coupling to climate and climate change is crucial when it comes to predictions of future variability of soil moisture storage and exchange with the atmosphere and vegetation. Long term data sets of measured soil moisture are of critical importance to achieve a better understanding of how these systems interact and to identify the main drivers for seasonal and long term soil moisture variations. Drought and feedback mechanisms between soil moisture and extreme temperatures are documented in the literature (Lanen et al., 2016; Perkins, 2015; Samaniego et al., 2018). Mass and energy fluxes in soils are coupled processes (Zehe et al., 2019). Due to less evaporative cooling during drought periods, temperatures tend to be higher (Hirschi et al., 2011). A review of soil moisture and climate interactions is given in Seneviratne et al. (2010).

Main drivers of soil moisture dynamics are rainfall (wetting) and the vegetation period (radiation driven drying) (Mälicke et al., 2020). Vegetation can influence the soil water budget through an increase in transpiration, hydraulic lift of water from lower soil layers, reduced runoff on steep slopes and reduced soil evaporation due to shading (Liancourt et al., 2012). Other im-

pact factors include soil type, local groundwater availability, inclination angle and direction of exposition (Schnellmann et al., 2010). Feedback mechanisms between soil moisture and groundwater resources with weather phenomena like El Niño are also described in the literature (Kolusu et al., 2019; Solander et al., 2020). The 2015-2016 El Niño event is associated with extreme drought and groundwater storage declines in Southafrica while at the same time in east African countries south of the equator

an increase in precipitation and groundwater recharge was recorded (Kolusu et al., 2019). Similarly, Solander et al. (2020) found evidence for both, increase (eastern Africa) and decrease (northern Amazon basin, the maritime regions of southeastern Asia, Indonesia, New Guinea) in soil moisture storage depending on location. An increase in catchment evapotranspiration was observed during the past decades (Duethmann and Blöschl, 2018). As groundwater recharge is dependent on the availability of excess soil moisture, therefore aquifers respond to climatic periodicities (Liesch and Wunsch, 2019).

Traditionally, soil moisture was determined by taking representative soil samples for gravimetric determination, following oven drying. The main disadvantage of this method, despite being very precise, is its destruction of the sampling location and the sample itself. Achievement of long term data sets is challenging using this method. Non destructive measurement methods include cosmic ray neutrons (Rivera Villarreyes et al., 2011; Kędzior and Zawadzki, 2016), installation of TDR sensors (Li et al., 2019), thermal infrared sensors (Yang et al., 2015), resistivity measurements like the OhmMapper (Walker and Houser,

2002), capacitance measurements or neutron probes (Hodnett, 1986; Evett and Steiner, 1995). A comparison and discussion of several sensor systems using different measurements principles is given in Jackisch et al. (2020), highlighting also the need for thorough calibration before the use of such systems. During this study two calibrated neutron probes were used. Numeric approaches include modeling of depth-dependent soil moisture based on surface measurements (Qin et al., 2018) or modeling of soil moisture for specific locations based on available weather data (Menzel, 1999). Another modeling approach of soil

moisture is based on remote sensing data. This has been done on catchment scale (e.g. Pellenq et al., 2003; Penna et al., 2009), regional scales (e.g. Mahmood et al., 2012; Otkin et al., 2016; Long et al., 2019) and the global scale (eg. Dorigo et al., 2017; Albergel et al., 2019) with various calculation grid sizes and temporal resolutions. Analysis of inherent parameter uncertainty in modeled soil moisture and implications for current discussions about soil moisture dynamics should be considered (Samaniego et al., 2012), as well as upscaling of measurements to different temporal and spatial scales (Mälicke et al., 2019).

Lysimeters are also suitable for gaining in-depth knowledge about water balance and water movements in soil, which is the main reason the lysimeter in this study is operated. It provides a direct measurement of percolation rates through the soil, which makes it suitable for monitoring and demonstration of equivalency of the earthen landfill cover (Abichou et al., 2006). Application of lysimeters, however is not restricted to monitoring of legally acceptable percolation rates, but also allows for studies into water and solute transport in the vadose zone that would not be possible by other means (Singh et al., 2018). Their

usage allows for precise determination of evapotranspiration (ET), if soil water storage is accounted for to well below the root zone (Evett et al., 2012) as well asdetermination of incoming water at the land surface due to precipitation and non-rainfall-events like dew or fog (Groh et al., 2018). Furthermore, they are used for determination of preferential flow (Schoen et al., 1999; Allaire et al., 2009), particle transport (Prédélus et al., 2015) and contaminant transport in the vadose zone Goss et al. (2010).

There are about 2500 lysimeters installed at around 200 sites across Europe, around half of them in Germany (Sołtysiak and Rakoczy, 2019). In the present study, we analyze long term soil moisture time series from two large inclined lysimeters located in southern Germany. Data from the monitoring of this test site has previously been evaluated and published concerning the proper function of the landfill cover (Zischak, 1997; Gerlach, 2007) and with regard to flow processes on steep hillslopes (Augenstein et al., 2015) using only much shorter parts of the time series available.

However, a time series analysis of all available soil moisture measurements from this test site to gain insight into long term soil moisture variations has not been done previously and is the main focus of this study. The inclusion of previously unpublished data from the more recent soil moisture measurements allows for a more comprehensive analysis of the time series. Using the available data from this test site, it is possible to identify past events that led to significant changes in the long term dynamics of soil moisture. Main research questions are:

– How did the measured soil moisture levels change over the past decades?

– Can these changes be described by simple linear models, or does it require more sophisticated modeling approaches?

– Can exceptional hydro-meteorological events that had a lasting impact on soil moisture levels be identified as tipping points by statistical methods?

– Are there seasonal differences? During which time of the year did the greatest change in soil moisture level occur?

– Which part of the soil is affected the most?

## 2   Study site

The study site is located in southern Germany (8.337°N, 49.019°E) near the city of Karlsruhe (Fig. 2). The climate in the region is classified as warm temperate, fully humid with warm summers or as Cfb according to the Köppen-Geiger Classification scheme (Beck et al., 2018; Kottek et al., 2006). Mean annual precipitation is 760 mm (1990 – 2007, DWD station 2522, Karlsruhe). Annual precipitation and temperatures are shown in Fig. 1. Exceptionally dry years within the observation period between 1994 and 2019 are 2003 with 566.3 mm and 2018 with 566.7 mm of precipitation. Highest annual precipitation was recorded in 2002 with 981.6 mm, followed by 2013 with 972.4 mm of precipitation. Mean annual temperature was highest in 2018 (12.33 °C) and lowest in 1995 (9.69 °C).

Two large inclined lysimeters are embedded in the municipal landfill site Karlsruhe-West for mandatory monitoring purposes. Cross sections of both lysimters are shown in Fig. 3. The first lysimeter (Field 1) was built in 1993 and started operation at the end of that year. With a width of 10 m and a length of 40 m, it has a size of 400 m². The mean inclination angle is 23.5° (43.5 %) with a southern exposition. The recultivation layer (RL) in this field has a thickness of 100 cm, it is underlain by a drainage layer (DL) with a thickness of 15 cm followed by a mineral clay liner (MCL) and capillary barrier.

The second lysimeter (Field 2, pictured in Fig. 4) was built in 2000, with first measurements being taken in December of that year. It consists of two separate fields with a size of 10 m by 20 m each, resulting in a total size of 400 m². The mean inclination

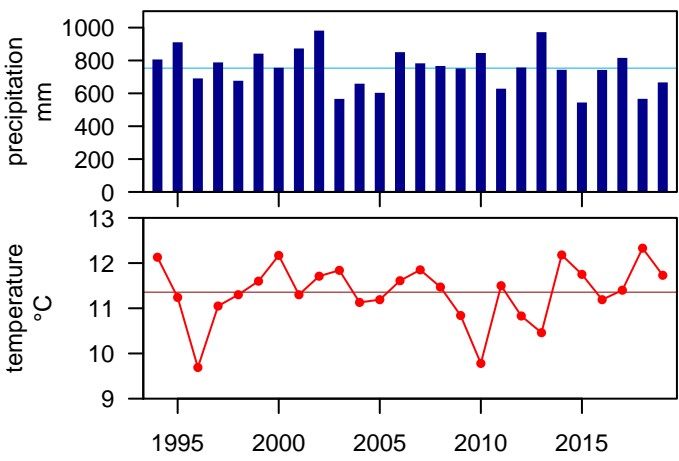

**Figure 1.** Precipitation and Temperature at stations 2522 (Jan 1994 – Oct 2008) and 4177 (Nov 2008 – Dec 2019) (DWD Climate Data Center (CDC), 2020)

angle is 23.5° (43.5 %) with southern exposition. Results from Field 1 showed, that a thicker RL is necessary in order to protect the MCL from drying out. This insight was considered during the construction of Field 2. The RL in Field 2A has a thickness of 200 cm, in Field 2B it has a thickness of 215 cm. It is underlain by a drainage layer (DL) with a thickness of 15 cm followed by a mineral clay liner and capillary barrier. Depth across the inclined field varies. Additionally, the mineral clay liner is not

present in the lower half of Field 2B, reducing the final depth of the lysimeter by 50 cm, affecting measurements taken at NP5 and NP6 below the RL. The RL was constructed by adding layers of soil on top of the compacted surface of the previous layer. Use of different materials in the soil layers can not be ruled out. Further details on the construction of both fields is given in Augenstein et al. (2015). The soil properties of the RL relevant to this study have been modeled by Gerlach (2007) using HELP (Berger, 2015). For the year 2002, the porosity of the RL is 0.4 [-], usable field capacity 0.25 [-] and the wilting point at

0.14 [-]. The permeability was estimated as $k_f = 1.0 \cdot 10^{-6}$ [ms$^{-1}$]. Formation of preferential flow paths in the lysimeter lead to changes in hydraulic properties over time (Gerlach, 2007).

Both fields are covered by grass and weeds, depending on the current season. The growth of deeply rooted plants that would damage the sealing system is prevented and the grass is cut regularly on the complete landfill cover including both lysimeters. In recent years, sheep have been used to limit the growth of vegetation in a more natural way. Further records on the vegetation

cover and plant maintenance are not available.

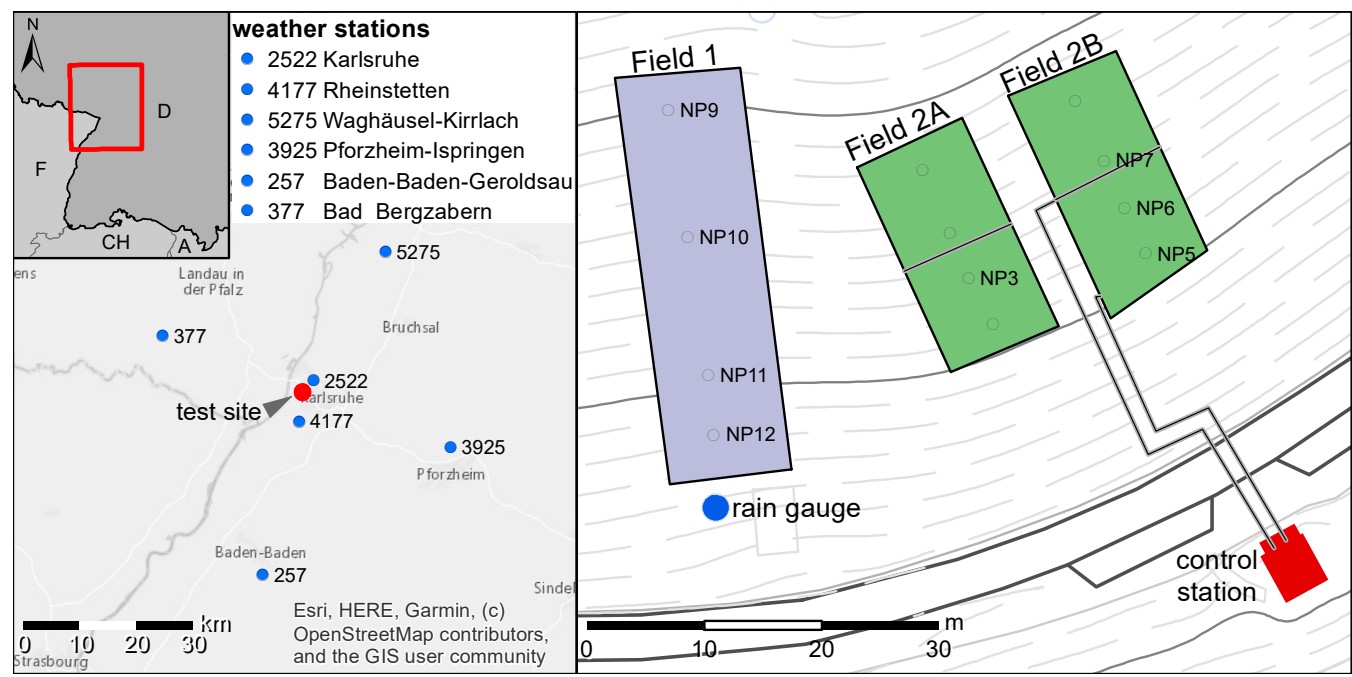

**Figure 2.** Location of the study site on a municipal landfill site in Karlsruhe, Germany, and locations of the weather stations used in this study. Lysimeter 2 consists of two separate fields (Field 2A, Field 2B).

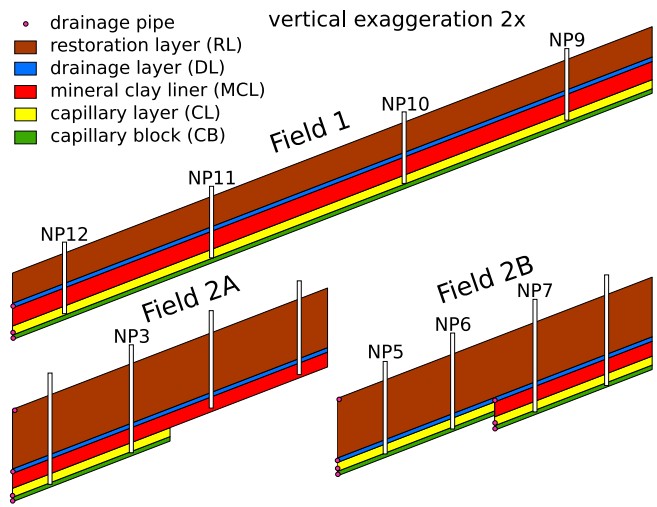

**Figure 3.** Cross sections of lysimeter Field 1 and Field 2 with the different layers and moisture measurement points.

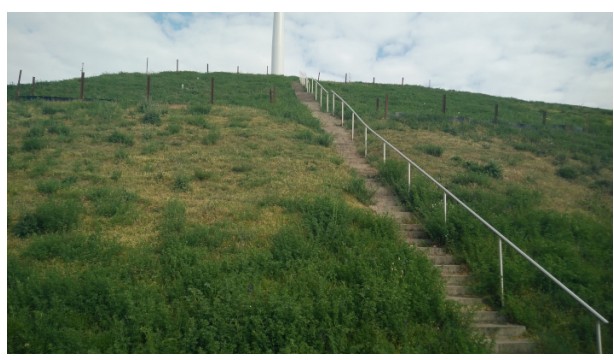

**Figure 4.** Lysimeter Field 2 (visible in the upper part of the image between vertical beams)

## 3 Material and Methods

### 3.1 Soil moisture and discharge measurements

Soil moisture measurements were carried out using two different neutron probes. A modified Wallingford IH2 neutron probe was used until 23 August 2012. From 30 November 2012 onward, a modified Troxler 4300 Depth Moisture Gauge was used.
Both models used an Am/Be source with activities of 1.85 GBq and 370 MBq respectively (Augenstein et al., 2015). They were modified to fit into the installed measurement tubes. Selected measurement points are shown in Fig. 2. Neutron probe measurement points (NP) are constructed from steel tubes (⌀ 40.5 mm) installed vertically in the soil column. At neutron probe measurement point 9 through 12 (NP9, NP10, NP11, NP12) located in lysimeter Field 1, measurements were carried out on a weekly basis until Field 2 was constructed (December 2000). After construction of Field 2, measurements were taken monthly
in Field 1. At the same time, weekly measurements in Field 2 at neutron probe measurement points NP3, NP5, NP6 and NP7 started. Measurements were taken in depth increments of 10 cm until the bottom of the lysimeter is reached (final depth Field 1: between 2.1 m and 2.3 m; final depth Field 2: between 2.8 m and 3.4 m). No measurements were taken at the remaining four points in Field 2. During the period of January 2014 to June 2014, no measurements were taken at neither of the two fields due to ongoing construction of new accessibility stairs for Field 2.

Discharge from the drainage pipes (Fig. 3) is collected in cylindrical tubes equipped with magnetic valves at the bottom. A data logger connected to pressure sensors in each tube records water levels at regular intervals. Additional data points are recorded when the changes in water levels are large. Once the tube is full, the valve at the bottom opens and and closes automatically to empty the tube.

From changes in the recorded water levels, discharge was calculated. The area of the lysimeter field was used to calculate
monthly aggregated discharge per area (mm) from the amount of water that was collected (L).

**Table 1.** Location of weather stations and distances to the test site.

| Station name | Station ID | Elevation | Latitude | Longitude | Distance |
|---|---|---|---|---|---|
| Karlsruhe | 2522 | 112 m a.s.l. | 49.0382° | 8.3641° | 2.9 km |
| Rheinstetten | 4177 | 116 m a.s.l. | 48.9726° | 8.3301° | 5.2 km |
| Bad Bergzabern | 377 | 210 m a.s.l. | 49.1070° | 7.9967° | 26.7 km |
| Pforzheim-Ispringen | 3925 | 333 m a.s.l. | 48.9329° | 8.6973° | 28.1 km |
| Waghäusel-Kirrlach | 5275 | 105 m a.s.l. | 49.2445° | 8.5374° | 29.0 km |
| Baden-Baden-Geroldsau | 257 | 240 m a.s.l. | 48.7270° | 8.2457° | 33.2 km |

## 3.2 Additional data

Additional data used for this study include daily precipitation and modeled values of usable field capacity (uFC). Daily precipitation data at a station near Karlsruhe is published by the German weather service (DWD) (DWD Climate Data Center (CDC), 2020). Data for this station (Station ID: 2522) is available for the time range until October 2008. Another station, still in operation by the DWD, is located in Rheinstetten (Station ID: 4177), approximately five km south of the test site, providing data from November 2008 onward. Locations of both weather stations are shown in Fig 2.

The DWD also publishes derived model results for usable field capacity (uFC) (DWD Climate Data Center, 2020) that can be used for comparison of measured soil moisture time series. They are provided for two different soil types and as depth resolved values for the top 60 cm of the soil column. They are computed by the agrometeorological model AMBAV. For this study the depth resolved values for soil moisture under grass with sandy loam (wilting point 0.13 [-], field capacity 0.37 [-]) were used. Additionally, soil moisture under grass and loamy sand (wilting point of 0.03 [-], field capacity 0.17 [-]) up to 60 cm depth was used. A defined constant water content is used as boundary condition at the bottom of the model. Further model input parameters are hourly values of temperature, dew point, wind speed, precipitation, global radiation and reflected long-wave radiation. Data was used from five stations (Tab. 1: 4177, 377, 3925, 5275, 257, Fig. 2) and covers a time range from 1 January 1991 until 31 December 2019. Values at station 3925 are available from 2005 onwards. Measured soil moisture data is not directly comparable to uFC, because of a different scale being used. The uFC of 100 % is defined as the soil moisture content that can not be drained by gravity. Nonetheless, both measured soil moisture and usable field capacity have similar temporal distribution patterns.

## 3.3 Theory and calculations

Volumetric water content ($\theta$) and uFC are expressed as %. Data analysis and visualization was done in the R system for statistical computing R Core Team (2020).

Time series were transformed into a radial coordinate system, to highlight the asymmetry of the seasonal cycle between gradual drying and fast re-wetting of the soil. New x- and y-coordinates for each measurement were calculated according to Eq. 1 and Eq. 2.

$$x = \cos\left(2 \cdot \pi \cdot \frac{d_{year}}{d/a}\right) \cdot \theta \tag{1}$$

$$y = \sin\left(2 \cdot \pi \cdot \frac{d_{year}}{d/a}\right) \cdot \theta \tag{2}$$

In these two equations, $x$ and $y$ are the new coordinates in a radially transformed coordinate system, $\theta$ is the volumetric water content in %. $d_{year}$ is the day of the year. It is divided by the length of the respective year ($d/a$) in order for $2\pi$ to equal one year.

Mean soil moisture of the recultivation layer in Field 2 was calculated as average of NP3 at depths between 10 cm and 180 cm and at NP5, NP6 and NP7 at depths between 10 cm and 220 cm.

For each individual depth, a linear regression was calculated using all measurements for the years 2000 to 2019 (see 3.1). Calculations were done using the `lm()`-function in the R system for statistical computing (R Core Team, 2020). As linear regression can be dependent on the selected start and end times, additional regressions were calculated over the complete

available time span, based on time series cut off before 2004 and between 2004 and 2016. The resulting slope of these regression lines represent the mean change of soil moisture in $\% d^{-1}$. A conversion into $\% a^{-1}$ was calculated by using the average length of 365.2425 $da^{-1}$, according to the Gregorian calendar.

To overcome the limitations of linear regressions when used on data with large seasonal variation compared to a small overall trend, another set of linear trends was calculated based on monthly averages. The monthly values were calculated as

averages based on the measured values within each month and depth. No weights were assigned to individual measurements. The time series for all depths were each subdivided into twelve time series, one for each month. For example, application of this subdivision on the time series at a depth of 170 cm at NP3 results in twelve time series (Fig. 5). Linear regressions were then calculated separately, based on all mean values for each month, giving the average slope for each measurement point, depth and month. An example for these calculations is shown in Fig. 5 for the months of April and September at NP3 and at a

depth of 170 cm.

Measurements at Field 1 were taken weekly at the start of the time series, but the interval changed to monthly measurements later. Therefore, use of all values for regression would lead to an over emphasis of the early part of the time series, due to the higher number of samples during that time span. To avoid this bias and over emphasis, monthly averages were used. The regressions yielded individual values for the change in soil moisture by month and depth. Additionally, further information

about the regressions were extracted from the results. These include standard deviations and p values for the slopes. The analysis was done with the time series of uFC in a similar fashion.

Time series analysis are sensitive to the selection of a suitable model. To overcome the paradigm of the single-best model approach in time series decomposition, Zhao et al. (2019b) implemented a Bayesian model averaging scheme to approximate

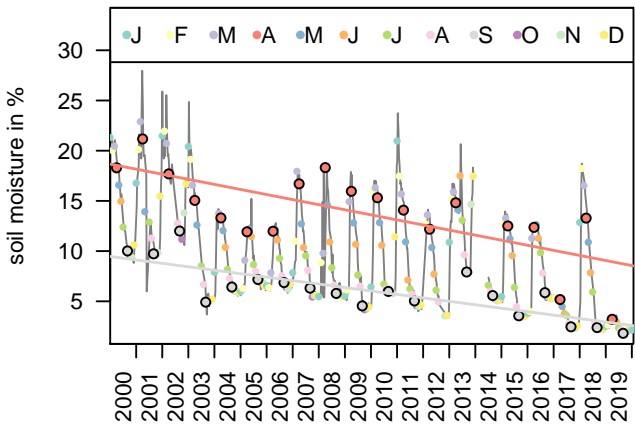

**Figure 5.** Example for the calculation of monthly linear regressions for April and September at NP 3 and at a depth of 170 cm.

complex relationships by the use of Markov Chain Monte Carlo stochastic sampling. The model space is explored by randomly traversing through combinations of coefficients. The number and location of individual changepoints in seasonality and trend are randomly sampled and all candidate models averaged based on how probable each of them is. Results of the model not only include best estimates for model parameters (e.g. location of changepoints), but also their probability distributions. Bayesian change point detection and time series decomposition was done using the `beast()`-function from the Rbeast package (Zhao et al., 2019a). This divides the time series into seasonal and trend components, along with change points in both. The period was set to 12 for monthly time series decomposition. The same monthly averaged time series were used as with the previous monthly linear regressions.

## 4 Results

The study represents a very specific case and the interpretation of results is limited to these specific conditions at the landfill.

Measured soil moisture values in the RL at NP5 and NP10 are presented in Fig. 6 at the corresponding position on the respective soil moisture profiles and before monthly averages were calculated. There is a gap in measurements during the first half of 2014. Field 2 was built in 1999 and no data is available prior to the year 2000. In total, over 140 000 individual soil moisture measurements are shown. Due to grain size and soil properties, the mineral clay liner has a higher moisture content (> 25 %). It is overlain by the gravel drainage layer, which has a very low moisture content. For evaluation, only soil moisture content of the RL is used in this study (n = 91198), because it is thought to be the layer in the lysimeter that reflects best the processes and moisture dynamics found in natural soils.

From this figure, a seasonal pattern is clearly visible. Soil moisture increases relatively quickly in late autumn or winter, especially in the upper parts of the soil. After reaching a critical soil moisture level, discharge from these layers starts more or less instantaneous and is measured as discharge from the DL. This wetting period is followed by a more gradual drying period, starting in late spring and lasting until the consecutive wetting period. The years before 2003 appear to have higher

soil moisture content and shorter drying seasons, especially at, but not restricted to, Field 2. This can be seen for example at NP5 in Field 2 where blue colors, indicating soil moisture of over 30 %, alternate with green colors (15 %) before 2003. After 2003, green colors alternate with yellow colors, indicating soil moisture below 10 %. In recent years, the re-wetting of the soil during the winter month repeatedly did not reach the lower parts of the soil, especially in Field 2. For example at NP5 in depths between 100 cm and 200 cm yellow colors indicate soil moisture levels below 5 % for the complete years 2017 and 2019. Measured discharge during these years was significantly lower compared to the prior years. Despite above average precipitation during the second half of 2017, re-wetting was only observed much later in early 2018. Precipitation in 2018 was well below average, and paired with a large atmospheric demand for ET, once again drying out the lower soil and no re-wetting occurring in the winter months.

Soil moisture in Field 1 is higher at the upper slope (NP9) compared to the lower slope (NP12), especially at the start of the measurement series (Fig. A1). As with Field 2, soil moisture levels are lower after 2003. Because the RL is not as thick in Field 1 (100 cm) compared to Field 2 (~215 cm), re-wetting in the lower soil in depths below 100 cm is not observable. However, in years with missing re-wetting (e.g. 2017, 2019), of lower soil in Field 2, a similar gap can be observed in Field 1 below the MCL (~200 cm). In data from Field 2, depth-dependence of soil moisture is clearly evident. Higher soil moisture at depths of around 100 cm sharply decreases over the next 20 cm and downward propagation of the moisture front is also delayed. This effect is caused by differences in soil compaction during construction of the lysimeter and possibly the use of different soil materials. The volume of the lysimeter was filled in several layers and soil consolidated in between each. Porosity and hydraulic conductivity is therefore not uniformly distributed over the complete depth of the lysimeter. Greatest differences are found at the interfaces of two consecutive stages of construction between strongly consolidated top of the underlying layer and the less densely packed bottom of the overlying layer. The consistent and very distinct break of soil moisture over the entire measurement period suggests that there is a distinct change in porosity and hydraulic conductivity between these two layers. Settling down of the soil cover in the years after construction may additionally change soil properties over time.

Values of modeled uFC are also shown in Fig. 6 for DWD station 4177 under grass and loamy sand.

## 4.1 Drainage data and discharge behavior

Discharge from the drainage layer is shown in Fig. 6 d and e. It follows a seasonal pattern with highest discharges being recorded at the beginning of the year, usually around the months of January or February. During the summer months, discharge is lowest and can be completely absent, especially in recent years. The onset of discharge is usually more or less instantaneous, with highest rates of discharge measured around the time discharge starts. Augenstein et al. (2015) analyzed the discharge behavior as a function of the soil moisture. It usually takes the soil moisture front several weeks to percolate through the soil column and eventually drain. Individual precipitation events during the drier summer months do not lead to an immediate onset of discharge from the lysimeter. However, precipitation events during the discharge period may rapidly increase the amount of discharge. The soil moisture near the bottom of the soil column at the initial onset of discharge is usually lower than the soil moisture when the discharge stops.

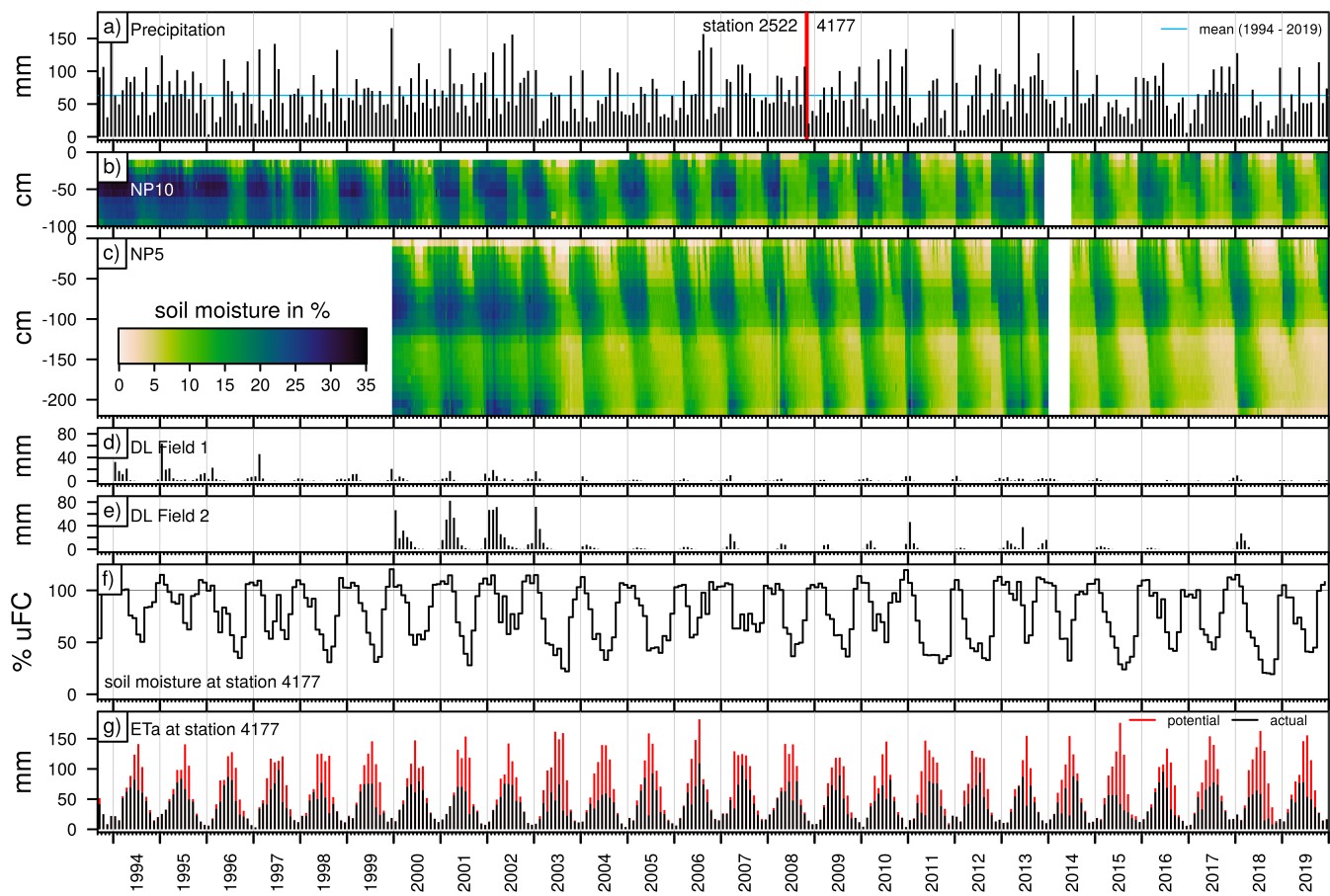

**Figure 6.** a) Monthly precipitation at stations 2522 (Jan 1994 – Oct 2008) and 4177 (Nov 2008 – Dec 2019) (DWD Climate Data Center (CDC), 2020). Blue line represents mean monthly precipitation during 1994 to 2019. b) and c) Time series of selected soil moisture measurements (NP5 and NP10 respectively) at the test site near Karlsruhe, Germany. d) and e) Monthly discharge of the drainage layer (DL) at both lysimeters. f) Simulated monthly averages of usable field capacity (loamy sand), g) Monthly values of simulated potential evapotranspiration (red) and simulated actual evapotranspiration (black) (DWD Climate Data Center, 2020) at station 4177. Measurements on Field 2 are available from 2000 onward. No measurements were taken during the first half of the year 2014.

Water flow is also influenced by the slope. The soil moisture front on the upper slope usually takes longer to reach equivalent depth on the lower slope. Meaning the lower slope usually gets wet faster, indicating a strong lateral component of sub-surface water flow (Augenstein et al., 2015).

Initially, after construction of the lysimeter, discharge was noticeably higher in comparison to later years. This is more pronounced in Field 2 (Fig. 6 e). A significant reduction in annual discharge from the DL can be seen around the year 2003. This reduction in drainage coincides with a reduction in soil moisture in both fields. In recent years when the soil moisture front did not reach the lower parts of the soil column, there was no discharge from the DL.

## 4.2   Asymmetry of drying and re-wetting

To highlight the asymmetry of the seasonal cycle between gradual drying and fast re-wetting of the soil, two exemplary time series are shown in a polar coordinate system (Fig. 7). For comparison, the soil moisture time series of NP3 at a depth of 170 cm and a mean time series from all sampling points at Field 2 are shown. The overall trend of both time series is quite similar, however the asymmetry is much more pronounced in the time series of NP3 at 170 cm. The mean of all soil moisture time series in Field 2 was calculated over the complete depth of the recultivation layer (RL). Due to the lag in the downward propagation of soil moisture in the profile, the asymmetry of the seasonal cycle is evened out by calculation of the mean soil moisture over multiple depths and measurement points.

In the two time series shown in polar coordinates, the graph based on mean values describes a circle for each year of observation. In the time series of NP3 at 170 cm on the other hand, the graph describes spirals resembling nautiluses for each year of observation. The decreasing radius over time, apparent from both time series, indicates a decrease in soil moisture. White areas between lines indicate large and sudden changes in soil moisture levels during especially dry years. The opening of the nautilus corresponds to a rapid increase in soil moisture during winter. Depending on precipitation conditions, this increase may occur at the end of the year or the beginning of the consecutive year.

## 4.3   Overall linear regressions

In Fig 8, results of individual linear regressions of soil moisture measurements are shown for each depth and measurement point. Over the period between 2000 and 2019, soil moisture decreases by $0.34 \pm 0.14$ $\%a^{-1}$ within the RL. The observed decrease is lowest in the first 20 cm of the soil column at both lysimeter fields. At the depth of 10 cm there is even a small increase observable in Field 2.

Overall, the decrease in soil moisture is most pronounced at depths of 20 cm to 40 cm in Field 1 (NP9, NP10). Due to the thicker RL compared to Field 1, highest absolute decrease is found at a greater depth of around 100 cm in Field 2. Below 130 cm at Field 2, absolute rate of soil moisture change is slightly lower. Seasonal variations of soil moisture patterns larger than the overall trend lead to a relatively low coefficient of determination ($0.20 \pm 0.10$). However, with exception of two points (NP5 20 cm, NP7 10 cm), all slopes of calculated regression lines are significant ($p < 0.05$, n = 122). Coefficients of determination are lowest at the top and increase until a depth of around 100 cm. Precipitation events lead to short term variations in soil moisture. These variations are larger at the surface.Downward movement of the water in the soil column is being dampened

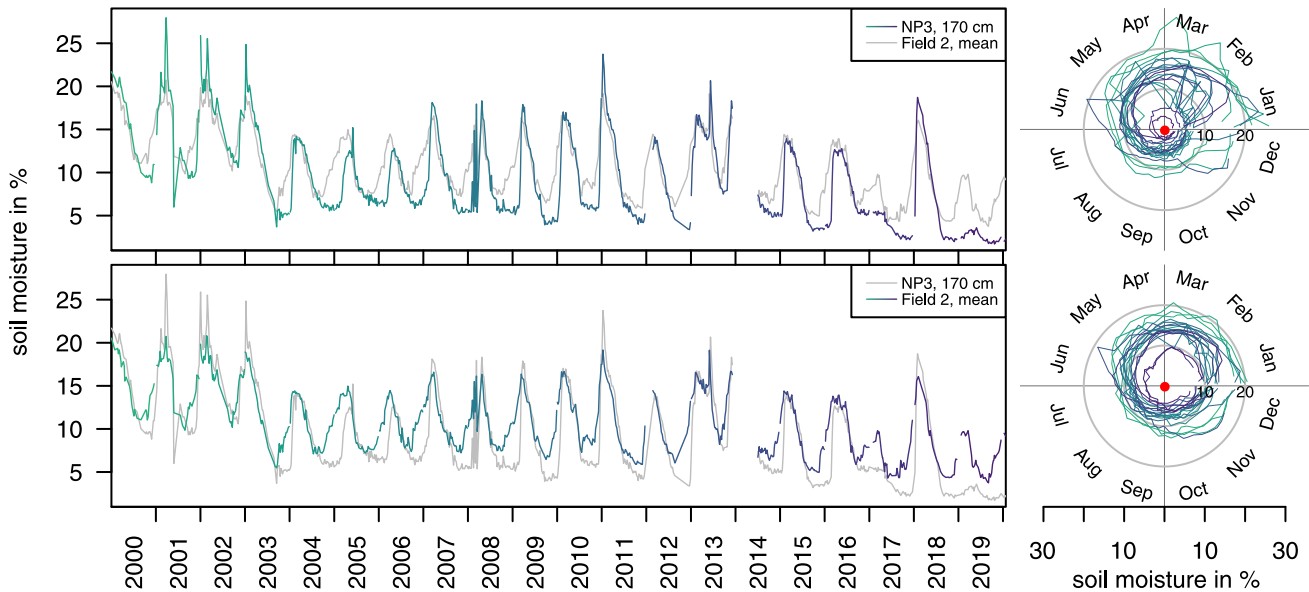

**Figure 7.** Time series of soil moisture at NP3 at 170 cm and mean soil moisture of Field 2 as well as the same data in a polar coordinate system to highlight seasonal asymmetry of gradual drying and fast re-wetting as well as the overall trend of declining soil moisture. Colors indicate the year of the measurement. For context, gray lines showing mean soil moisture of Field 2 and soil moisture at NP3 at 170 cm respectively were added.

with depth. At some depths, soil moisture patterns are more persistent. This might be due to different materials being used or differences in compaction during construction of the lysimeters and landfill cover. Differing soil properties like porosity, hydraulic conductivity and capillary forces determine the water retention capacity of the soil.

## 4.4 Monthly linear regressions

The results of linear regressions based on the monthly averages is shown in Fig. 9. Resulting slopes with $p > 0.05$ (i.e. soil moisture change is not significant) are indicated by a marker.

A statistically significant increase in soil moisture can be observed in the top 10 cm of Field 2 (NP3, NP5, NP6, NP7) during the winter months only. Most other values show a significant decrease in soil moisture. The moisture change in the top 60 cm of the soil does show an increase during summer, but this increase is not statistically significant. The lack of statistical
significance might be due to the shorter length of the time series at Field 2 compared to Field 1. As previously mentioned, overall soil moisture levels were higher before 2003. Inclusion of additional data before this point, as is the case with Field 1, would push the resulting decrease in soil moisture towards higher absolute values. From depths of around 70 cm to 130 cm (70 cm to 90 cm at NP3), decrease in soil moisture has a semiannual distribution. Highest reductions in soil moisture occurred during November and December as well as during April and May. Below this, decrease in soil moisture is generally lower and

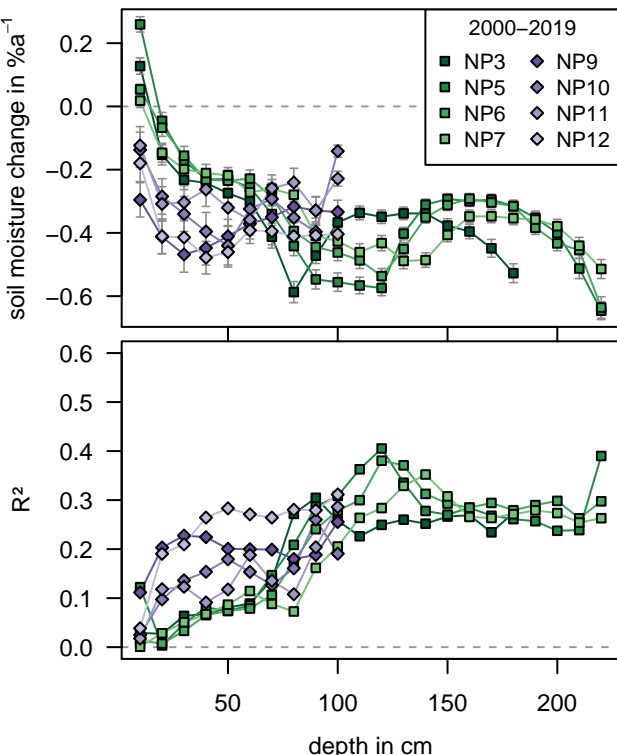

**Figure 8.** Results of individual linear regressions for soil moisture measurements in the recultivation layer, expressed as change in soil moisture content $\left[\%a^{-1}\right]$.

does show a weak annual cycle with highest values in December and January and minima during June and July. Highest values are shown in the lowermost 30 cm of the RL directly above the DL between January and May.

In Field 1, a decrease in soil moisture can be observed at all depths. The semiannual distribution of soil moisture change is similar to that of Field 2. It is most pronounced during spring and autumn and less pronounced during winter and summer.

The winter months are usually times of largest groundwater recharge and highest soil moisture in the lower soil. In recent years however, less water percolated through the upper parts of the soil at both lysimeter fields, affecting especially the soil moisture levels in the lower soil. This drying effect is amplified by the DL. It drains excess water and inhibits capillary rise. This means the depth of evapotranspiration in the lysimeter is greater than two meters and includes the complete RL.

Results of linear regressions based on monthly averages of uFC are shown in Fig. 10. Most values indicate a decrease of soil moisture, but at the same time, most linear regressions are not statistically significant ($p > 0.05$). However, results for station 5275 indicate a clear and significant decrease in soil moisture during most of the year. The decrease in the lower soil layers appears to happen later in the year. Compared to Fig. 9, the semiannual pattern is not as visible, but some months (August at station 4177, 377, 5275, 257) do show lower annual changes or even an increase in uFC (January at station 3925).

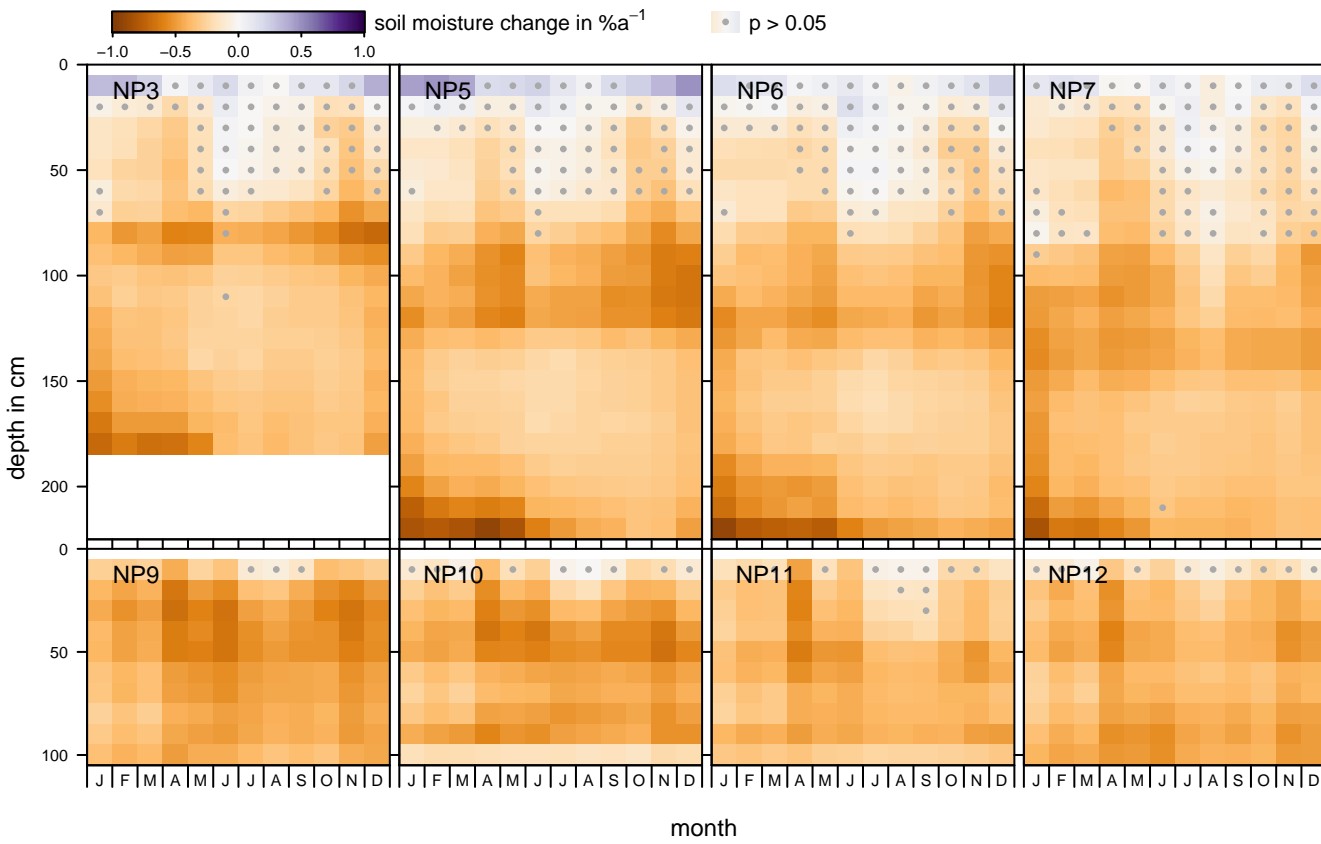

**Figure 9.** Results of individual linear regressions for soil moisture measurements in the recultivation layer, expressed as change in soil moisture content $\left[\%a^{-1}\right]$ calculated over the complete time series for each month based on monthly averages. Upper graphs: Field 2, lower graphs: Field 1. Values for p > 0.05 are indicated by a marker.

Compared to largest decrease in measured soil moisture at the beginning of the vegetation period in April and at the end of the vegetation period in November, the decline of uFC at the end of the vegetation period appears to happen much earlier (3925, 5275).

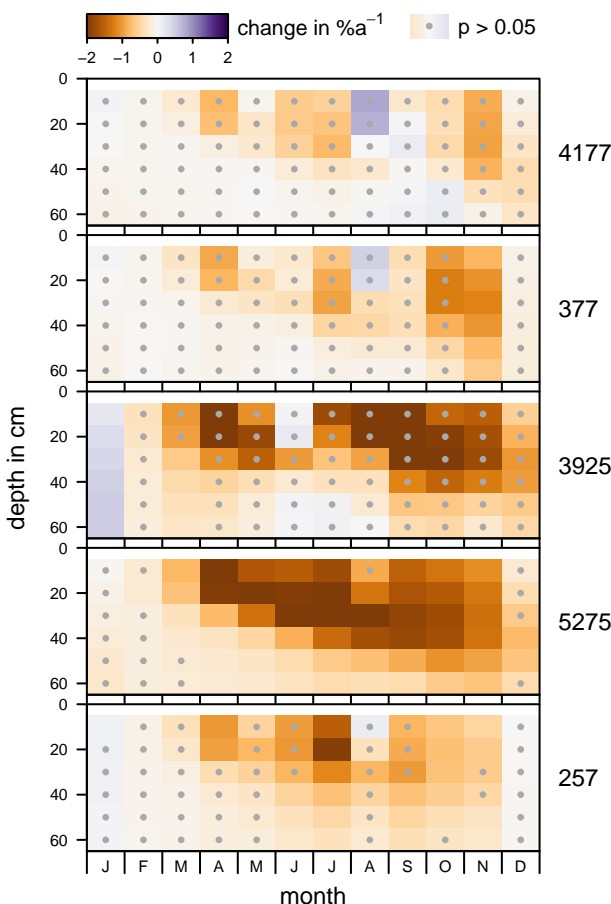

**Figure 10.** Results of individual linear regressions for usable field capacity (uFC) for the top 60 cm (at stations 4177, 377, 3925, 5275, 257), expressed as change in usable field capacity $\left[\%a^{-1}\right]$ calculated over the complete time series for each month based on monthly averages. Statistically non-significant values (p > 0.05) are indicated by a marker.

## 4.5 Time series decomposition

During modeling with Rbeast, the time series are decomposed into a trend component and a seasonal component, along with change points in both, seasonality and trend. Individual calculations are done for each depth increment at all measurement points. An example for NP3 at a depth of 170 cm is given in Fig. 11. The trend component shows a positive slope before 2003. A changepoint in trend with a probability of 68% was discovered in February of 2003. After another changepoint with a lower probability in December 2003 (17%) the soil moisture trend stabilized at a lower level after a significant decrease in soil moisture levels between February and December. Changes in seasonality were detected in 2004 and 2006/2007. In between these, the amplitude of the seasonal variations was lower.

In Fig. 12 the main results of this model are shown for a measurement point in Field 1 (NP5) and Field 2 (NP10). This kind of decomposition allows for easier visual analysis of the underlying trend component (Fig. 12). Probabilities of change point

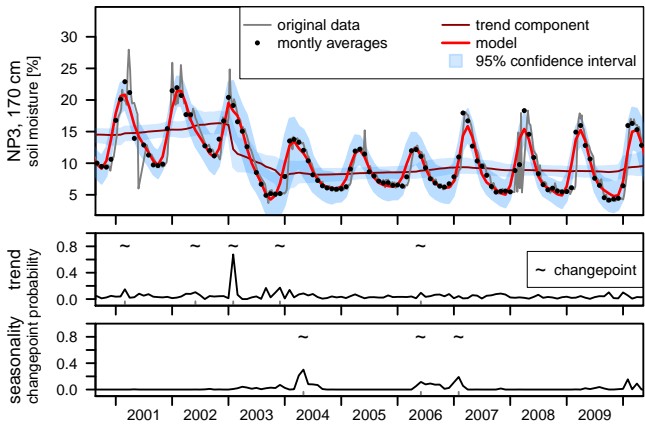

**Figure 11.** Results of time series decomposition for NP3 at a depth of 170 cm. Changepoints and their respective probability distributions are shown also.

occurrence indicate times of significant changes in trend and seasonality. Overall, higher soil moisture contents are apparent before 2003 and during a shorter time period in 2013/2014. In the past few years, soil moisture is noticeably lower, especially in depths below 100 cm.

The decomposed time series of NP10 in Field 1 (NP9 - NP12 in Supplement A5) reveal higher initial soil moisture contents, followed by a gradual decrease over time. The decrease is most pronounced at the beginning of the measurement series, until around 1998 a more or less stable level of soil moisture is reached. The amplitude of seasonality at the top of the slope (NP9 and NP10) during this time of high initial soil moisture is lower. This is probably due to the maximum saturation of the soil being reached, leading to an increase in discharge from the soil instead of further increase in soil moisture content and storage. In 2003, a change point in trend is visible. Modeling resulted in high probabilities for this change point. In the following years, the soil moisture is at a lower level. Apart from the elevated soil moisture before 2003, higher soil moisture is also evident in 2013. The distribution of probabilities for a change point in trend does not show a clear cut during this event. Probabilities are elevated over a wider range of time. The amplitude of soil moisture seasonality is more or less stable for the remainder of the time series and does not show high probabilities.

Measurements at Field 2 (NP 3, NP5, NP6, NP7) also show higher initial soil moisture contents. As previously mentioned, depth dependence of soil moisture due to lysimeter construction is also visible in the deconstructed time series. No apparent trend is observable until the year 2003. A change point in trend and the corresponding probabilities is then visible around the same time as in Field 1. In the following year (2004) a change point in seasonality with elevated probabilities in the lower half of the RL at Field 2 occurred. Slightly elevated probabilities for this change point were already calculated for the year 2003 itself. In general, amplitudes of seasonal variations are higher towards the top of the RL. After the 2003 change point, higher amplitudes of seasonal variation are found lower in the RL than before (NP3, NP5, NP6). At NP7 (Supplement A5), the

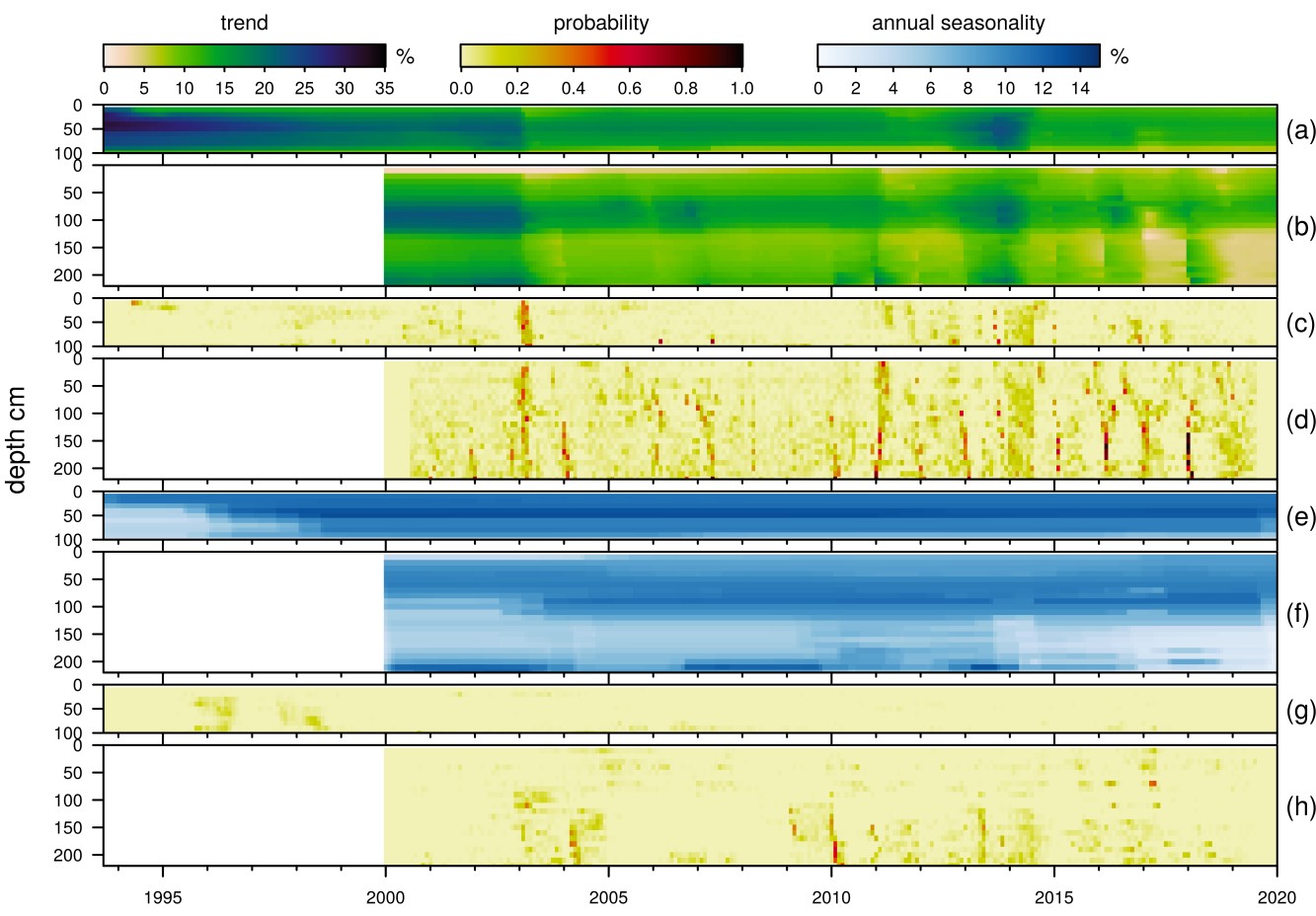

**Figure 12.** Results of modeling soil moisture at NP5 and NP10 with Rbeast. (a) Trend component of soil moisture time series at NP10 in Field 1. (b) Trend component of soil moisture time series at NP5 in Field 2. (c) Probability of change point in trend component at NP10 in Field 1. (d) Probability of change point in trend component at NP5 in Field 2. (e) Amplitude of annual seasonality derived from seasonal component at NP10 in Field 1. (f) Amplitude of annual seasonality derived from seasonal component at NP5 in Field 2. (g) Probability of change point in seasonality component at NP10 in Field 1. (h) Probability of change point in seasonality component at NP5 in Field 2.

amplitude of seasonal variations at a depth of 80 cm increased after this point, but amplitudes in the soil below are significantly lower.

Another visible change point in trend with elevated probabilities is visible at the end of 2011. This change point cannot be seen in Field 1. After 2015, change points with elevated probabilities appear to occur almost every year. At the same time,
reduction of soil moisture to a low level not observed previously occurs, mainly in the lower half of the RL. Because of a thinner RL, this effect cannot be observed in Field 1. In recent years from 2015 onward, amplitude of seasonal variation in the lower half of the RL is greatly reduced, because dry soil without the reoccurring annual re-wetting does not show significant seasonality.

Interruption of capillary rise due to lysimeter construction inhibits re-wetting of the lower soil from groundwater. Thus,
results of this study might not be applicable to soils with a shallow depth to the groundwater surface or modeled values of usable field capacity. Boundary conditions are different for the modeled usable field capacities analyzed in this study. They are calculated from weather data and standardized soil properties. An additional source of soil moisture is provided by capillary rise due to a constant moisture boundary condition at the bottom of the model. The fact that some stations did show the same patterns as measured soil moisture, while other stations with same soil properties did not, could mean, that there are feedback
mechanisms between soil moisture and the input parameters of the uFC model. Future studies should concentrate on these interconnections between soil moisture, groundwater recharge and groundwater level to determine if they amplify or dampen the temporal dynamics of soil moisture.

## 5   Discussion

One possible explanation for the rapid change in soil moisture levels could be a change in soil properties (water retention,
preferential flow paths, hydraulic conductivity, soil structure, etc.) as a result of singular extreme events like the exceptionally dry year 2003. Augenstein et al. (2015) found, that there are hysteresis effects during drying and re-wetting of the soil at this site. Water bound in different states (e.g. adhesive water or water stored in the inter layers of clay minerals) has different migration times. The proportion of water bound in these different states therefore influences the drainage behavior (Šimůnek et al., 2003). During the period of increased soil moisture, water migrates into the inter layers of the clay minerals. (Schnetzer,
2017) This water can not be drained by gravity but still contributes to soil moisture. After discharge from the lysimeter stops, desiccation of these clay minerals may occur by evaporation into the soil air.

Another contribution factor might be changes in soil temperatures (Vanderborght et al., 2017; Schneider et al., 2021). These are usually highest around September and October. Temperature has great effect on viscosity of water and influences surface tension and contact angle. Thus determining how much water can be retained by capillary forces.
Hydraulic conductivity in the vadose zone is dependent on the moisture content. This feedback mechanism might amplify or dampen the hysteresis, depending on the proportions of bound soil moisture in different states (available pore water, pore water in enclosed cavities). Furthermore, extreme drying of the soil might lead to non-reversible desiccation of clay minerals or formation drying cracks as preferential flow paths, both leading to significant changes regarding the overall hydraulic function-

ing of the whole system. However, though these likely phenomena may occur, changes in soil water dynamics are also visible
from the modeled uFC. These are not based on physical measurements which are dependent on time-constant soil properties, but rather use time constant properties of a model soil. The fact that these modeled values also show changes in their temporal soil moisture patterns gives ample evidence that the change points found are not merely a function of soil properties but of the local climate as well, which the modeled values are solely based on.

Changes in measured soil moisture at around the year 2003 could also be the result of the establishment of a vegetation cover after the construction of the lysimeter and over several consecutive years. The soil cover is important to prevent erosion and to lower overall percolation by increasing evapotranspiration. The system is designed with a vegetation cover as an integral part to it's proper functioning. However, Field 1, which has been constructed several years prior to Field 2, shows a similar change at the same time.

A similar change is likewise visible in the modeled data, but a change in vegetation cover is not used as an input to the model. It is still possible that vegetation and evapotranspiration both drive these changes in the model and the measured data, but then it has to be connected through the meteorologic parameters used in the model (e.g. longer vegetation periods). Ionita et al. (2020) found that prevailing large-scale atmospheric circulation may impact atmospheric blocking over the north sea and central Europe and thus lead to extreme weather to be more persistent. If this is the case, the change towards elevated temperatures would also lead to an extension of the vegetation period. Thus increasing evaporation as a result of higher temperatures, and plant transpiration as a result of the longer vegetation period. If evapotranspiration is limited by the amount of available water, the difference between actual evapotranspiration and potential evapotranspiration will increase. Less evaporative cooling and lower ambient humidity will then increase temperatures even further, increasing the severity of a drought.

Robinson et al. (2016) found evidence for the existence of drought induced alternative stable soil moisture states. They observed a step change that occurred at the beginning of 2004 with an apparent transition to a new stable state in which soil moisture levels never reached saturation again. They found water retention characteristics to change due to a loss of organic material by increased organic matter mineralization under moderate drought conditions. According to their findings, the bottom boundary behavior was modified from a seepage face behavior before 2004, to free drainage after. For arid regions, strong positive feedback between vegetation and soil moisture has been described by D'Odorico et al. (2007). Small changes in environmental variables can lead to rapid and irreversible shifts between two alternate stable states (D'Odorico et al., 2007).

## 6  Summary and conclusions

Aim of this study was to identify long term variations of soil moisture patterns and to identify the occurrence of particular events that led to tipping points in soil moisture levels. To achieve this, we analyzed high resolution soil moisture measurements from a test site near Karlsruhe, Germany. The data consists of depth-resolved, weekly soil moisture measurements in increments of 10 cm to a final depth of around 200 cm. Additionally, modeled data was used for comparison and interpretation of the results.

Over the investigation period, there is a significant decrease in soil moisture. This decrease is most pronounced at greater depths up to around 200 cm. Comparison of the measured soil moisture with modeled data of uFC for different stations indicates spatial heterogeneity, meaning future changes in soil moisture will vary in severity based on location.

The model depth of 60 cm is sufficient only when looking at the overall dynamics of uFC. Measurements of soil moisture at depths of up to two meters show significant seasonal variations well below the depth of the model. This large seasonal evaporation depth means, change in soil moisture storage at these depths are an important component in future climate change models that can not be neglected and further real world measurements are needed in order to calibrate these models.

Times of largest changes to the soil moisture levels are the beginning of the vegetation period in April and the the end of the vegetation period in November. This indicates that changes in the vegetation cover might be the large driver of the observed depletion of soil water.

Bayesian modeling of the soil moisture data revealed change points in both trend and seasonality that had high probabilities. It seems reasonable to suggest that specific events of extreme drought had a lasting impact on soil moisture storage and led to deep desiccation of the soil. The most pronounced tipping point being the one during the exceptionally hot drought year 2003. After this point, soil moisture levels were on a lower level. In recent years, soil moisture levels declined even further, accompanied by a decline in the amplitude of seasonal variations. Thus, the impact of a decline in soil moisture is not limited to absolute level of the overall trend, but includes a decrease in seasonality. The overall dynamics are changed without any sign of a return to the previous state. This change in seasonality can not easily be described by simple linear models. Further application of the data and conclusions presented in this study can potentially be used in a much wider context when applied to numeric modeling of soil moisture, vegetation and climate as well as their interactions.

*Acknowledgements.* We would like to thank A. Hoetzel from the AfA Karlsruhe. We thank all persons who worked on this project during the past three decades, doing maintenance on the measurement systems and the lysimeters, conducting the measurements with the neutron probe, recording and curating the data, and thereby creating a rich treasure of data which we could base this work on. We thankfully acknowledge the funding we received from the city of Karlsruhe during the monitoring program. We thank Gary Witherall for the speedy language editing. We acknowledge support by the KIT-Publication Fund of the Karlsruhe Institute of Technology.

**CRediT author statement:**

Markus Merk: Project administration, Data Curation, Formal analysis, Visualization, Writing - Original Draft

Nadine Goeppert: Supervision, Resources, Writing - Review & Editing, Funding acquisition

Nico Goldscheider: Supervision, Resources, Conceptualization, Writing - Review & Editing

**Data availability**

The data that support the findings of this study are available from the authors upon reasonable request.

## Competing interests

The authors declare that they have no conflict of interest.

## Financial support

We thankfully acknowledge the funding we received from the city of Karlsruhe during the monitoring program.

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

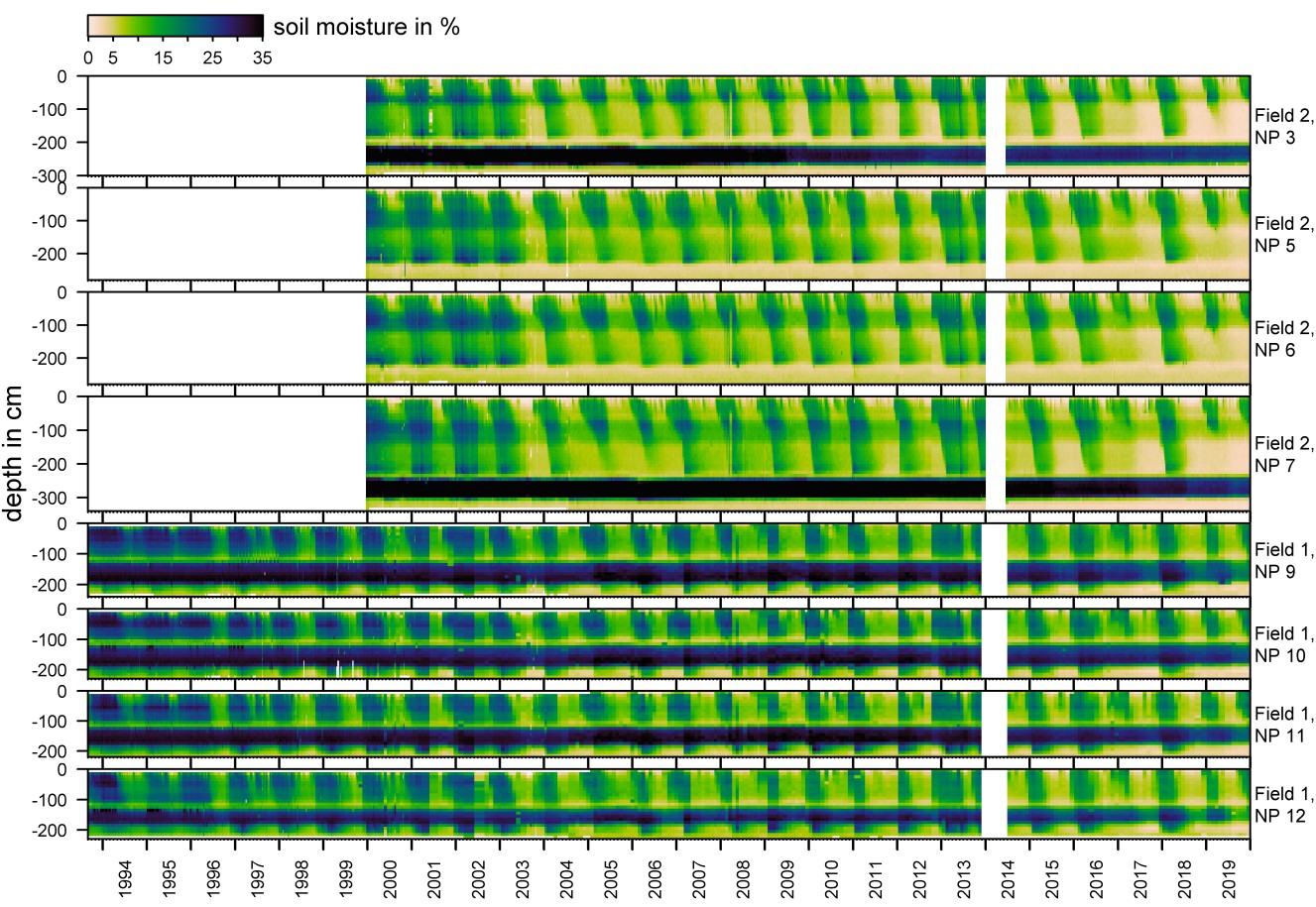

**Figure A1.** Time series of soil moisture measurements at the test site near Karlsruhe, Germany. Measurements on Field 2 are available from 2000 onward. No measurements were taken during the first half of the year 2014.

Zischak, R.: Alternatives Oberflächenabdichtungssystem "Verstärkte mineralische Abdichtung mit untenliegender Kapillarsperre". Wasserbilanz und Gleichwertigkeit, Dissertation, Universität Karlsruhe, Karlsruhe, 1997.

# Appendix A: Supplemental figures

610

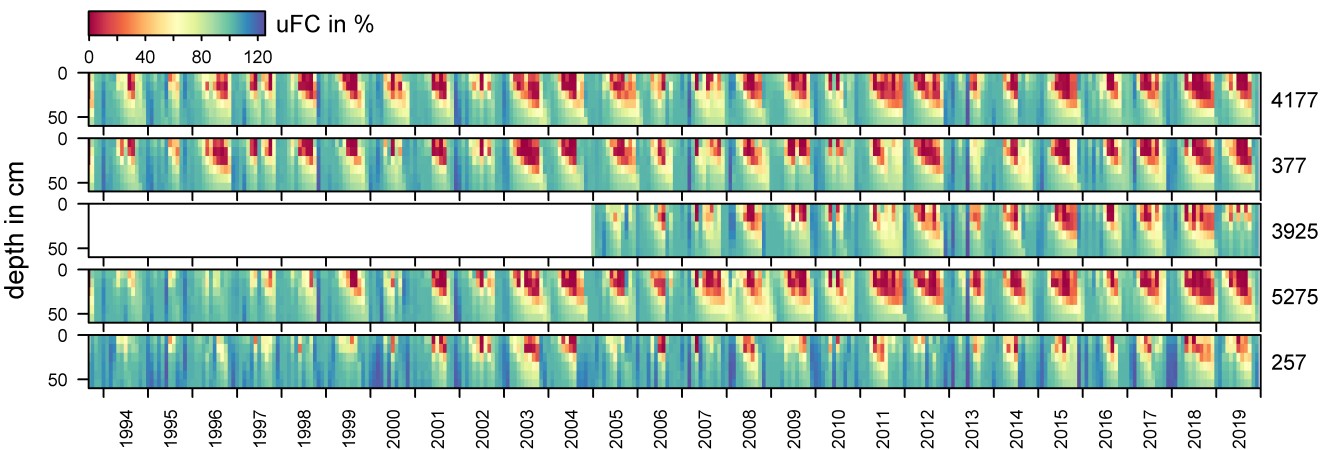

**Figure A2.** Monthly averages of usable field capacity calculated at five selected weather stations (DWD Climate Data Center, 2020). Values were computed by the agrometeorological model AMBAV. The model calculates soil moisture under grass with sandy loam. The soil sandy loam has a wilting point of 13 volumic% and a field capacity of 37 volumic%. Further model input parameters are hourly values of temperature, dew point, wind speed, precipitation, global radiation and reflected long-wave radiation.

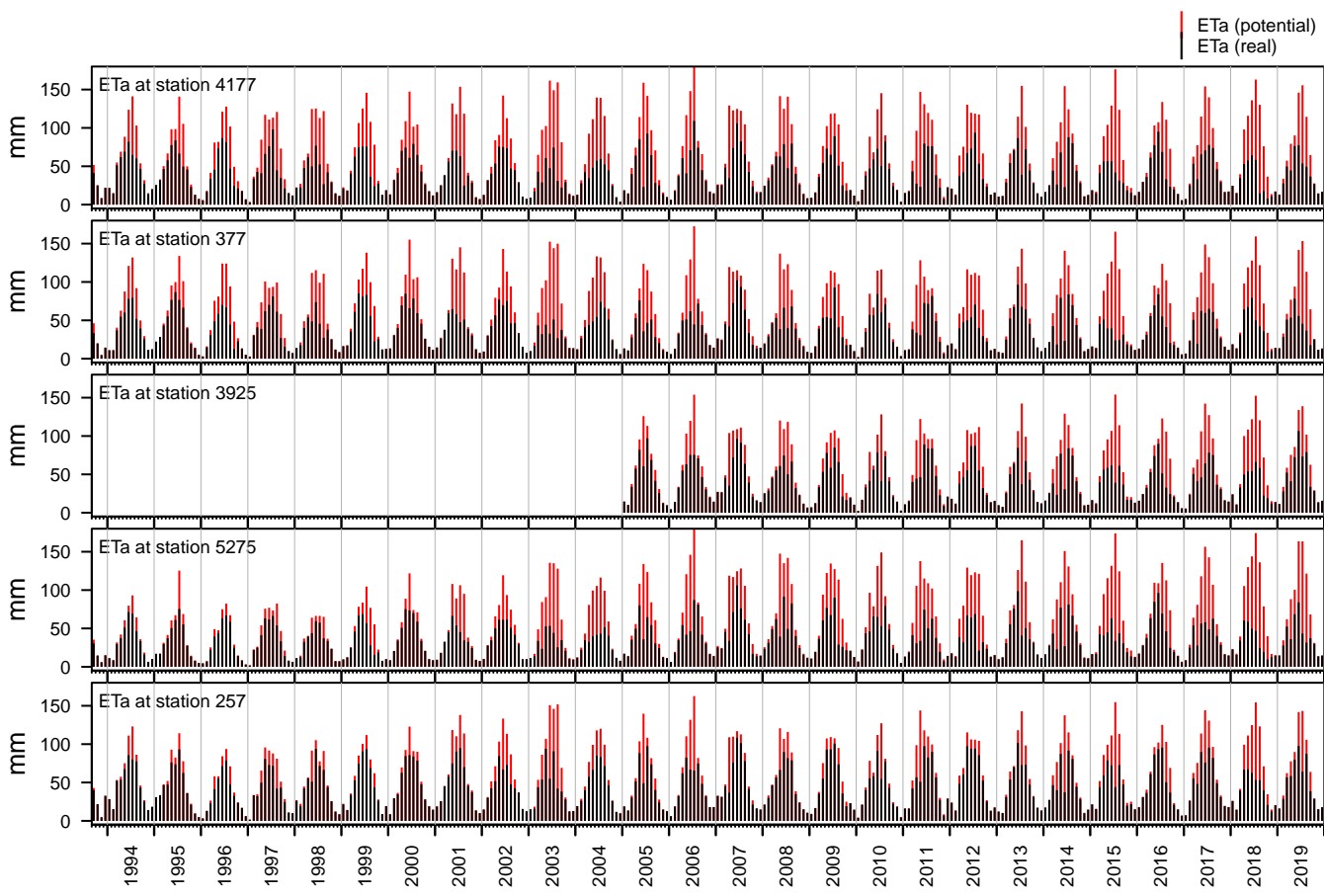

**Figure A3.** Potential and real evapotranspiration at five selected weather stations (DWD Climate Data Center, 2020).

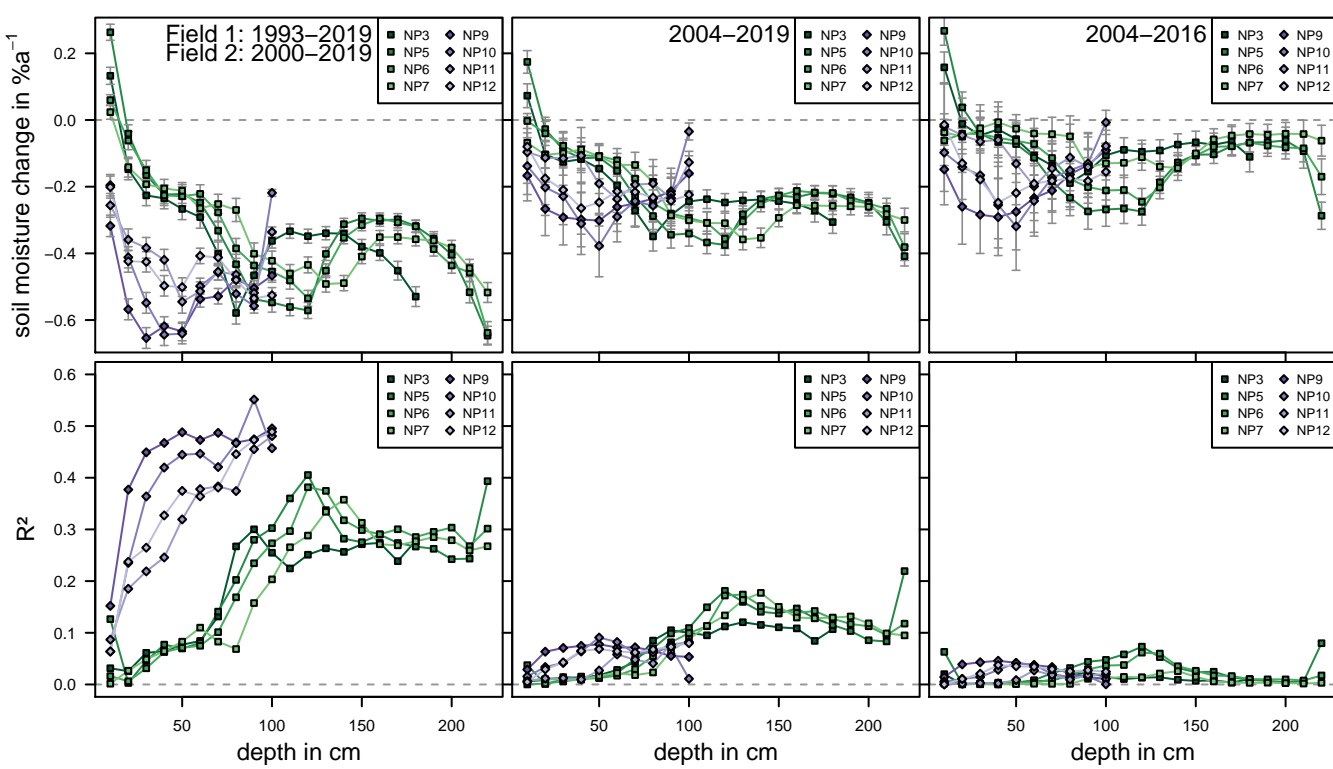

**Figure A4.** Results of individual linear regressions for soil moisture measurements in the recultivation layer, expressed as change in soil moisture content $\left[\%a^{-1}\right]$.

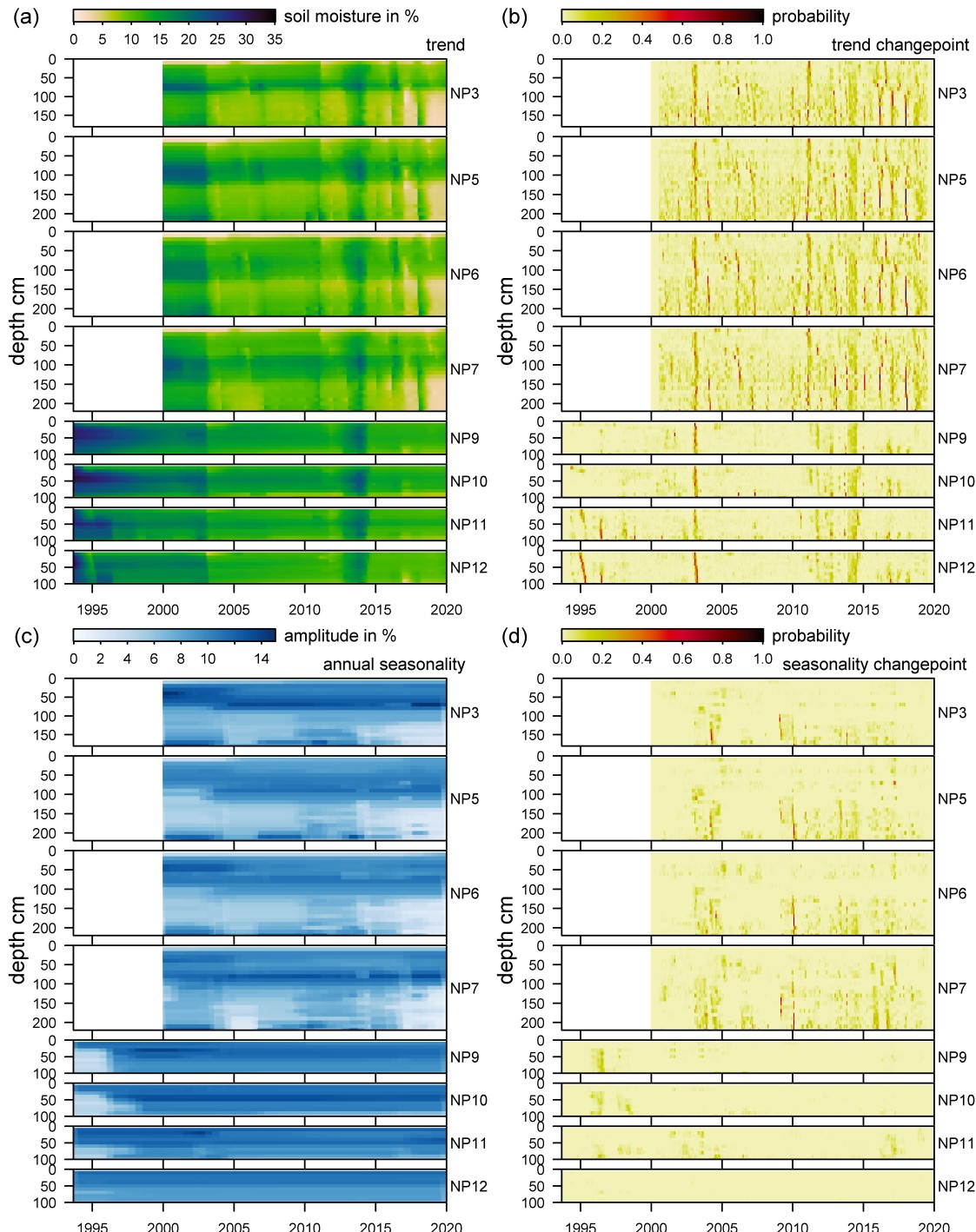

**Figure A5.** Results of modeling soil moisture with Rbeast. a) Trend component of soil moisture time series. b) Probability of change point in trend component c) Amplitude of annual seasonality derived from seasonal component. d) Probability of change point in seasonality component.

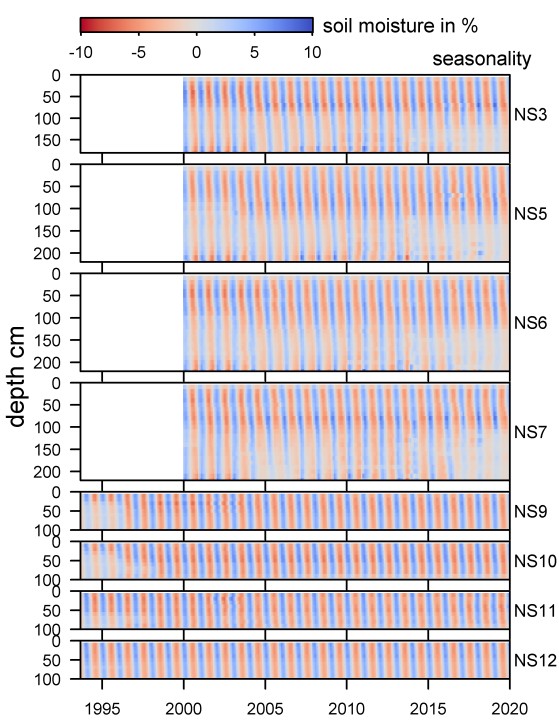

**Figure A6.** Seasonal component of soil moisture time series.