# Peer review of "Deep desiccation of soils observed by long-term high-resolution measurements on a large inclined lysimeter"

_Hydrology and Earth System Sciences, 2020_

## Referee Comment (RC1) · Katrin Schneider (Referee) · 3 Aug 2020

General comments The paper presents soil moisture data observations covering 20 years (and for some of the data even longer) measured at different soil depths and at a high temporal resolution. Soil moisture was measured at several locations in a large lysimeter on a landfill. The long and more or less continuous soil moisture observations with a high temporal resolution cover different climatic conditions and make the study valuable. This allows analyzing soil moisture development in the soil horizon under different climatic conditions over several years, including very dry conditions. The structure of the manuscript is clear and concise. Despite this wealth of data, I have

severe concerns that the way the data are presented and discussed misses important aspects. The discussion and interpretaion remains rather vague and should go into more detail. The manuscript requires a major revision. My suggestions are listed below:

Specific comments

1. The manuscript does not provide basic information on factors that have substantial influence on soil water movement and evapotranspiration. Please provide these information in the study site description, and also use these additional data in the interpretation and discussion of your results (e.g. by applying an analysis of variance) a) The data stem from two large lysimeters installed at a landfill. This makes a very specific case study, since the soil layers have been build up artificially. This specific case is not discussed in the paper, but it seems as if the landfill cover can be compared to surrounding non-artificial soil or landscapes. Unfortunately, there is no presentation of the soil profile(s) of the two lysimeters and no description of soil properties, like texture, bulk density, pore volume, pF values and so on. I assume that the cover of the landfill has to meet specific requirements, and I would expect that information on soil properties therefore are available. I recommend including a table with information on soil properties (in different depths or discretized by the layer type, e.g. recultivation layer, drainage layer etc.) in the site description, and along with that, a figure of the soil profile with indications of compaction horizons or other information which are specific to that soil.

b) Along with missing information on soil properties, there is no description of the vegetation cover of the landfill (if there is a cover, or is it bare soil / something else?). If the two lysimeters have the same (vegetation) cover type, the effects on evaporation, transpiration and drainage are likely comparable. The second lysimeter was implemented later than the first one. Are there changes in the (vegetation) cover between the two? Please add this information in the study site description, and also consider it in the further discussion and interpretation of results.
c) A photo might be helpful to give the reader an idea of the site

2. Information on mean annual and monthly precipitation and temperature at the study site or in its vicinity (e.g. from DWD station data) is missing. Since the authors discuss the effects of the very warm and dry summer 2003 on subsequent soil moisture, it would help the reader to see some information on average conditions and on the deviation from long-term averages during the observation period. Please add a table and/or figure with mean annual and mean monthly precipitation and temperature at least, and indicate the deviation from these average conditions during your observation period. You may also consider to highlight years with very strong deviations (e.g. very warm/cold, wet/dry).

3. The methods section (3.3. Theory and calculations) is very brief. Especially the Bayesian change point detection should be explained in more detail. Please also add a reference to the software you used for calculating the linear regression models.

4. Comparison of soil moisture measurements with modelled uFC: a) When using the (modelled) usable field capacity (uFC) provided by DWD I wonder why you did not try to make these data more comparable to the volumetric soil moisture measurements from the two lysimeters. This could either be done by converting the modelled uFC into volumetric soil moisture making use of the soil properties (in particular pF values, pore volume) these calculations are based on - as far as these information are provided along with the modelled uFC data. Or do it the other way round and calculate uFC at the soil moisture sampling points based on the volumetric soil moisture content and the soil properties (e.g. layer specific pF values) of the soil layers of the two lysimeters. This touches the above-mentioned missing information on soil properties. b) A discussion on how well modelled uFC can be compared with soil moisture measurements at the the lysimeters is missing. Presumably, the modelled uFC is based on non-artifical soils, but soil moisture observations at the two lysimeters represent conditions in an 'artificial' soil layer. Please include a more detailed discussion here, or skip the modelled uFC data, if the soil properties on which the calculations are based are not comparable the

conditions at the lysimeters.

5. Presentation of results: Figure 2 (discussed chapter 4) is hard to read and the information might therefore not reach the reader. Since there is many data 'squeezed' into this figure I find it hard to read or to really compare the different NP measurements. It is particularly difficult for the NP data of Field 1. Can think of another way of presenting the data, or (this goes more to the Editor) place this figure in landscape format? It might also be worth to plot it in a different way, e.g. calculate the difference from average for each depth increment over the entire observation period for each pixel/time step. I suggest to remove the map with modelled uFC at the bottom of Fig. 2 completely, or to move it to the appendix.

L. 149: explain climatic conditions 2003 (dry and hot summer in the study region) – this can be accompanied or underlined by further general information on climate characteristics at the study site over the study period (see also #2 of my general comments above)

L. 161 – 165: give a more detailed description on soil properties, and discuss the effects of soil compaction. Could a compaction horizon result in a capillary barrier in the soil layer? How would that effect soil water movement?

L. 193 – 199: please provide a more in-depth discussion in this paragraph on potential effects of soil compaction and why moisture patterns at some depths are more persistent than in other depth. E.g. continuously 'wet' conditions at approx. 100 cm in field 2 or at roughly 150 – 200 cm at field 1; why are there drier conditions in a small area above 150 cm at field 1).

L. 203 – 206: In this paragraph you argue that the shorter observation period in field 2 is reason why the observed decrease in soil moisture is not significant. I wonder if this is the only way of interpreting this result. A) When looking at Fig. 5 there seems to be a change in the direction of soil moisture change at approx. 70 cm at field 2. This might also correspond to the rooting depth (in case vegetation is present), resulting in

a quick recycling of precipitation via root water uptake / evapotranspiration which does not allow percolation to deeper soil layers. B) A compaction layer might further impede percolation. C) The different soil depths of field 1 and 2 and the different duration since the lysimeters have been installed, resulting effects on soil properties and (vegetation) cover should be discussed, too.

L. 209 – 212: please provide a more detailed discussion on the reasons for the observed reductions in soil moisture (e.g. in the context of precipitation / temperature regimes). Why is the reduction in the lowest part of the soil profile most pronounced from January – May in field 2, and why is it not obvious in field 1?

Figure 7, time series decomposition: this analysis is valuable to detect trend changes. As with Figure 2 I am concerned that, with the amount of data, the figure is still readable. As suggested before, it might be worthwhile to test different colour schemes, or highlight particularly relevant results.

L. 262: it is the first time in the manuscript that re-wetting from groundwater is mentioned. Are the lysimeters connected to the groundwater in this specific setting on a landfill? If so, please also describe this in the study site description

L. 270: which soil properties do you think of? Please include that in more detail in this paragraph

Technical comments:

Chapter 1

L. 27-28: which were the effects of El Nino? Please explain the results of the cited studies (Solander, Kolusu) in more detail (1-2 sentences) or skip it

L. 30-36: you could state which measurement type is used

L. 44-46: rearrange order of sentences (e.g. start with second sentence in this paragraph

L. 55 'with regard' instead of 'in regard' Please check and correct where necessary throughout the manuscript: - 'depth' and 'depths' - 'In depht' vs. 'in-depth' - 'at depth of' vs. 'at a depth of'

Chapter 2

Include information on soil properties (at least those that are most relevant for soil moisture movement/storage), (vegetation) cover of the landfill, and climate characteristics

Explain why the two lysimeters have different soil depths

L. 78: add year '...being taken in December 2000' / '.. in December of that year'

Chapter 3

Are there more neutron probe measurement points in the lysimeters (since NP numbers start with 3, and if so, why was that data not used

L. 90: when was Field 2 constructed?

L. 122 – 127: could you please explain more clearly what you did here?

L. 134 – 137: please describe in more detail the Bayesian change point detection and time series decomposition: how is it done and which information does it provide?

Chapter 4

L. 141: please indicate the mentioned clay layer in Fig. 2

L. 155 – 157: it is hard to see the discussed results in the Figure in its current form (see #5 in general comments)

L. 157: change '...the missing occurrence...' into a less complicated sentence

L. 160: delete 'or so'

L. 168: delete duplicate 'the bottom of'

L. 170: depending on the DWD station, there is an annual cycle of uFC at 60 cm depth, so I would not call it 'minimal'

L. 171: 'at a depth'

L. 172-173: Why is there a clear change in modelled uFC already after 2001, but for the soil moisture measurements this is only visible after 2003?

L. 177: add 'and a mean time series from all sampling points at field 2 are shown'

L. 178 and 183: change 'in the individual time series' to 'in the time series of NP at 170 cm'

L. 190: change 'could be observed' to 'soil moisture decreases by 0.34 % . . .'

Figure 3: even with a good colour printer it is difficult to discern the colours representing the different years in the polar coordinate system. I suggest to try out other colour or gray scale palettes, or you just highlight very dry or very wet years.

L. 202: change end of sentence to 'are indicated by a marker'

L. 208: I would not use 'bias' in this case, since the differences are not artefacts

L. 215 – 218: please discuss in more detail why less water is percolating during winter in recent years

L. 226: typo in the DWD station code?

L. 228: 'increase' instead of 'increases'

---

## Referee Comment (RC2) · Jannis Groh (Referee) · 11 Aug 2020

The manuscript presents an interesting topic and shows long-term measurements of soil moisture at a municipal landfill site in Germany. The data covers a relatively large period with quite distinct climatic conditions (wet, dry years) and measurements on soil moisture are available for several depths and various profiles at the site. Despite the rich data set I found it difficult to read, because of the uncomplete description of the experimental data and the used methods in this study. The unclear description of the lysimeter/field cover and drainage data of each filed makes it difficult to interpret the results. The authors should include at least information on the land surface cover and

their change or development over time. The same should be done for the drainage data and interactions between climate, vegetation, and soil should be investigated and discussed. Hence I recommend the authors to include more data and consequently explore their data more in deep! In addition the authors should include a discussion of their results in the context of other study in the Results and Discussion section. Nevertheless it was an interesting read and I want to encourage the authors to carefully rewrite, revise and improve their manuscript.

Specific comments:

L13: Soil moisture is not a flux. Please reformulate the sentence.

L28: Change evaporation to evapotranspiration.

L31-36: Che authors may include also the discussion about soil moisture measured by different method see e.g. Jackisch et al. (2020).

L39-40: Change eg. to e.g.

L49:High precision and high temporal resolved measurement with lysimeter are also able to exactly determine incoming water at the land surface due to precipitation and non-rainfall-events like dew or fog. I suggest to add this point here.

L53: Not clear if observations from one or two lysimeters are used in this study? Please clarify this point.

L56-67: At this stage it is not clear to me why the authors need lysimeter in this study. Neither the introduction text nor the objectives are link to lysimeters. Please clarify this point! If not any soil profile with long term soil moisture measurements could be taken here instead of a more sophisticated measurement set-up with lysimeters.

L73-82: Please provide a table with detailed info on soil properties for all fields. This includes not only the basic info on soil texture but also other important information's, which are normally available at such municipal landfill experimental sites.

L85-86: The authors should explain the modification of the measurement devices in detail, if not study results might not be comparable to other studies.

L93: How does this number fit with the mean inclination angle of 23.5° for each field reported in L75 & L79?

L102: The authors should explain how the used uFC data were derived for this study. This includes the important assumption e.g. vegetation, model, boundary conditions and soil types/ properties.

L102-107: Totally unclear why the authors want to use the model uFC? This should be explained in the section.

L103: Which soil types are used for this uFC data? The authors should describe the soil properties to be able to compare it with the landfilled soil profiles.

L109-116: The authors should clarify why time series were transformed into a radial coordinate system. This was done only for soil moisture observations?

L117: Explain more in detail why the authors used linear regression and for what. Did the authors also check if the assumptions for using such a model are full filled?

L121: What happens in leap years? Why using 365.2425 instead of just using the length of the corresponding year, which can be 365 or 366 days long!

L134-137: Please explain the used method i.e. "Bayesian change point detection" and "time series decomposition" used in this investigation more in detail! The used methods should also be included in the introduction section and it should be shown why this kind of methods are appropriate for such an investigation.

L139: Please be precise: Figure 2 shows the monthly soil moisture profiles at the corresponding position, which were derived from the single measurements. I recommend also to add a) to the soil moisture and b) to the uFC subplot. I suggest in addition to use the same coloring scheme for both subplots. This makes it easier for the reader to

compare between soil moisture and uFC.

L142: Please explain why only RL will be evaluated?

L152: The authors should show this recharge data for each lysimeter in a separate figure! In addition to that the authors might show the precipitation data in the figure. Please discuss the different conditions during the observation period e.g. dry years, wet years and its implications for the observed soil moisture.

L153: Re-wetting 2018. This might be related to the in general wetter conditions in 2017! The authors should explore their data more in depth!

L139-165: The general patterns of the soil moisture can be seen relatively well from the figure 2. However, other results discussed here are difficult to see from this figure. I suggest the authors to re-think what the main purpose for showing the figure here is and change it in a way that main findings are clearly visible.

L139-165: Please discuss results in the light of soil properties at each plot and the vegetation of these fields/lysimeters.

L168: Please report which model bottom boundary was by the DWD to simulate uFC and discuss this in the light of the presented uFC values.

L171: Not sure from which observations I can see evaporation depths over 200cm from Figure 2? Please explain your findings more in detail! Augenstein et al. (2015) reports that the fields are covered by grass, so the authors should discuss also here the vegetation development of the lysimeters/fields and refer in the manuscript consequently to evaporation and transpiration. Was there any change in the vegetation over the observation period? From my perspective higher soil moisture values at the beginning of the period might be rather related to the establishment of vegetation on the fields i.e. change from bare soil with only evaporation to a field cover with grass including evaporation and transpiration. Please clarify this point!

L171: After looking at Augenstein et al. (2015) the authors should also clarify in the

M&M section that the depths across the inclined field varied. In addition the authors should include the info that layers of the profiles e.g. in field 2B are not the same missing mineral clay liner (referring Fig. 1b in Augenstein et al. (2015))

L176: For a better comparison of the time series I recommend to put both in on plot. This makes it easier to compare.

L177: Mean soil moisture of what depths or measurement profiles?

L175-181: It would be interesting to see this time series for the field 1 as the time series of this plot is much larger than for field 2.

L175-181: First: I can see from the time series specific changes after the extremely dry year 2003. Please discuss here possible reasons! You might have a look at e.g. Robinson et al. (2016) or Groh et al. (2020), which showed within their investigations a change in the soil moisture level after drought events. Rahmati et al. (2020) showed for two grassland site a trend of decreasing seasonal minimal soil moisture after drought event in 2015. I guess there is much more literature on that point and I suggest the authors to include a more profound discussion/comparison of their findings.

L139-187: I recommend the authors to use additional methods to analyze the soil moisture time series. It would be worth also to include time series of precipitation and potential evapotranspiration. The authors could also look at the relations between those variables and soil moisture observations e.g. by Wavelet-analysis (see e.g. Graf et al., 2014; Bravo et al., 2020; Rahmati et al., 2020).

L-Figure 2: The authors should explain visible artefacts, i.e. strange lines between 2007 and 2008, white points, and strange lines in 2004 [. . ..].

L188 & 200: Did the authors check the important assumptions associated with a linear regression model? The questions arises as I can see a change in soil moisture level after drought event in 2003, which might affect the distribution of the data.

L198-199: Please reformulate the sentence.

L201: Please show also the values for field 1 below 100 cm in figure 5.

L205-206: Please explain in detail why data for field 1 before 2003 where excluded here.

L207-208: Not sure why the inclusion of data before 2003 would bias the results of field 1?

L215: The authors should show and discuss this percolation data.

L215-218: The authors should clarify to which field this results are related.

L219-229: The authors should as already mentioned provide the background info of this model simulation in the M&M section. This is important to better understand and especially discuss the results. So e.g. which vegetation was used in the simulation, which model, does this model provide a coupling of plant and soil dynamically or use of a fixed LAI and so on.

L227: Unclear how the authors come to this conclusion. Please clarify this!

L230: I could not fully evaluate this section as there is very few information on the used methods in the M&M section.

L235: The authors should explain why 2003 was that important for the soil moisture and actually discuss reason for this observations. Please do this in the whole manuscript.

L238: Is this related to climatic conditions, evolution of the land surface cover or due to changes in the soil after packing the lysimeter? Please clarify this point! The authors also should be aware that landfill soils might behave different than natural developed soils.

L262: Yes indeed that might be a reason! This is actually the first line where discussion of the results starts! However I want to point out that current lysimeters might overcome such issues as those systems are able mimic not only a more dynamical recharge but also the capillary rise from shallow groundwater or deeper soil layers. For further

details on this lysimeters see Unold and Fank (2008); Pütz et al. (2016); Herbrich et al. (2017); Groh et al. (2020); and the effect of shallow groundwater table on land surface water fluxes Kollet and Maxwell (2008); Groh et al. (2016).

L263: This is not truth as the model used by the DWD accounts also for capillary rise. See https://www.dwd.de/DE/fachnutzer/landwirtschaft/dokumentationen/allgemein/bf_erlaeuterungen.pdf;jsessionid=C443840 at the chapter "Hintergründe zum Modell".

L271: I could not find any data in the manuscript that actual shows that hysteresis plays are role at this site. So please clarify the following sentence: "There are clearly hysteresis effects during drying and re-wetting of the soil".

L278-280: Very vague statement! Please discuss this in a broader context and compare findings with other studies!

L300: Not sure if the observation provide the info if this processes are irreversible or reversible! Please discuss this before in the Results & Discussion section.

L301: I am confused about this statement as the authors used a simple linear model in this manuscript!

L301-303: That's truth! Thus I recommend the authors to include also vegetation and drainage data to further explore their already rich data set and to include possible interactions between land surface cover, soil moisture and drainage.

Augenstein, M., Goeppert, N., Goldscheider, N., 2015. Characterizing soil water dynamics on steep hillslopes from long-term lysimeter data. Journal of Hydrology, 529: 795-804, https://doi.org/10.1016/j.jhydrol.2015.08.053. Bravo, S., González-Chang, M., Dec, D., Valle, S., Wendroth, O., Zúñiga, F., Dörner, J., 2020. Using wavelet analyses to identify temporal coherence in soil physical properties in a volcanic ash-derived soil. Agricultural and Forest Meteorology, 285-286: 107909, https://doi.org/10.1016/j.agrformet.2020.107909. Graf, A., Bogena, H.R., Drüe, C.,

Hardelauf, H., Pütz, T., Heinemann, G., Vereecken, H., 2014. Spatiotemporal relations between water budget components and soil water content in a forested tributary catchment. Water Resources Research, 50(6): 4837-4857, 10.1002/2013WR014516. Groh, J., Vanderborght, J., Pütz, T., Vereecken, H., 2016. How to Control the Lysimeter Bottom Boundary to Investigate the Effect of Climate Change on Soil Processes? Vadose Zone Journal, 15(7): 1-25, 10.2136/vzj2015.08.0113. Groh, J., Vanderborght, J., Pütz, T., Vogel, H.J., Gründling, R., Rupp, H., Rahmati, M., Sommer, M., Vereecken, H., Gerke, H.H., 2020. Responses of soil water storage and crop water use efficiency to changing climatic conditions: a lysimeter-based space-for-time approach. Hydrol. Earth Syst. Sci., 24(3): 1211-1225, 10.5194/hess-24-1211-2020. Herbrich, M., Gerke, H.H., Bens, O., Sommer, M., 2017. Water balance and leaching of dissolved organic and inorganic carbon of eroded Luvisols using high precision weighing lysimeters. Soil and Tillage Research, 165: 144-160, 10.1016/j.still.2016.08.003. Jackisch, C., Germer, K., Graeff, T., Andrä, I., Schulz, K., Schiedung, M., Haller-Jans, J., Schneider, J., Jaquemotte, J., Helmer, P., Lotz, L., Bauer, A., Hahn, I., Šanda, M., Kumpan, M., Dorner, J., de Rooij, G., Wessel-Bothe, S., Kottmann, L., Schittenhelm, S., Durner, W., 2020. Soil moisture and matric potential – an open field comparison of sensor systems. Earth Syst. Sci. Data, 12(1): 683-697, 10.5194/essd-12-683-2020. Kollet, S.J., Maxwell, R.M., 2008. Capturing the influence of groundwater dynamics on land surface processes using an integrated, distributed watershed model. Water Resources Research, 44(2): W02402, 10.1029/2007WR006004. Pütz, T., Kiese, R., Wollschläger, U., Groh, J., Rupp, H., Zacharias, S., Priesack, E., Gerke, H.H., Gasche, R., Bens, O., Borg, E., Baessler, C., Kaiser, K., Herbrich, M., Munch, J.-C., Sommer, M., Vogel, H.-J., Vanderborght, J., Vereecken, H., 2016. TERENO-SOILCan: a lysimeter-network in Germany observing soil processes and plant diversity influenced by climate change. Environmental Earth Sciences, 75(18): 1-14, 10.1007/s12665-016-6031-5. Rahmati, M., Groh, J., Graf, A., Pütz, T., Vanderborght, J., Vereecken, H., 2020. On the impact of increasing drought on the relationship between soil water content and evapotranspiration of a grassland. Vadose Zone Journal, 19(1): e20029,

10.1002/vzj2.20029. Robinson, D.A., Jones, S.B., Lebron, I., Reinsch, S., Domínguez, M.T., Smith, A.R., Jones, D.L., Marshall, M.R., Emmett, B.A., 2016. Experimental evidence for drought induced alternative stable states of soil moisture. Scientific Reports, 6: 20018, 10.1038/srep20018. Unold, G., Fank, J., 2008. Modular Design of Field Lysimeters for Specific Application Needs. Water Air Soil Pollut: Focus, 8(2): 233-242, 10.1007/s11267-007-9172-4.

---

## Author Comment (AC1) · 16 Sep 2020

**RC1**

General comments The paper presents soil moisture data observations covering 20 years (and for some of the data even longer) measured at different soil depths and at a high temporal resolution. Soil moisture was measured at several locations in a large lysimeter on a landfill. The long and more or less continuous soil moisture observations with a high temporal resolution cover different climatic conditions and make the study valuable. This allows analyzing soil moisture development in the soil horizon under different climatic conditions over several years, including very dry conditions. The structure of the manuscript is clear and concise. Despite this wealth of data, I have severe concerns that the way the data are presented and discussed misses important aspects. The discussion and interpretaion remains rather vague and should go into more detail. The manuscript requires a major revision. My suggestions are listed below:

Specific comments

1. The manuscript does not provide basic information on factors that have substantial influence on soil water movement and evapotranspiration. Please provide these information in the study site description, and also use these additional data in the interpretation and discussion of your results (e.g. by applying an analysis of variance) a) The data stem from two large lysimeters installed at a landfill. This makes a very specific case study, since the soil layers have been build up artificially. This specific case is not discussed in the paper, but it seems as if the landfill cover can be compared to surrounding non-artificial soil or landscapes. Unfortunately, there is no presentation of the soil profile(s) of the two lysimeters and no description of soil properties, like texture, bulk density, pore volume, pF values and so on. I assume that the cover of the landfill has to meet specific requirements, and I would expect that information on soil properties therefore are available. I recommend including a table with information on soil properties (in different depths or discretized by the layer type, e.g. recultivation layer, drainage layer etc.) in the site description, and along with that, a figure of the soil profile with indications of compaction horizons or other information which are specific to that soil.

More Information on the lysimeters was added to the manuscript along with the available information on vegetation and soil properties. We added cross sections of both lysimeters. The cover was built as an alternative landfill cover not following or using any of the approved sealing systems at the time. Therefore the lysimeter was built to prove the proper functioning of the sealing system.

b) Along with missing information on soil properties, there is no description of the vegetation cover of the landfill (if there is a cover, or is it bare soil / something else?). If the two lysimeters have the same (vegetation) cover type, the effects on evaporation, transpiration and drainage are likely comparable. The second lysimeter was implemented later than the first one. Are there changes in the (vegetation) cover between the two? Please add this information in the study site description, and also consider it in the further discussion and interpretation of results.

We agree that comprehensive information on used materials is important in interpreting results. Unfortunately, detailed properties of the soil are not available. The major point of the monitoring program and reason for building of the lysimeters has been and still is the proper functioning of the landfill cover to stop water from percolating through the landfill itself. The material used as recultivation layer was only of minor importance

during construction. Overall the soil is very heterogeneous, containing clay, sand and even larger rocks.
We added available information on vegetation to the manuscript. Both lysimeters have the same vegetation cover consisting of grass and weeds. Unfortunately, not much information on soil properties and the establishment and past development of a grass cover is available.
c) A photo might be helpful to give the reader an idea of the site
We agree, photo added:

[Figure]

2. Information on mean annual and monthly precipitation and temperature at the study site or in its vicinity (e.g. from DWD station data) is missing. Since the authors discuss the effects of the very warm and dry summer 2003 on subsequent soil moisture, it would help the reader to see some information on average conditions and on the deviation from long-term averages during the observation period. Please add a table and/or figure with mean annual and mean monthly precipitation and temperature at least, and indicate the deviation from these average conditions during your observation period. You may also consider to highlight years with very strong deviations (e.g. very warm/cold, wet/dry).
We added a figure showing annual precipitation and mean annual temperatures to the manuscript. These data are sourced from the DWD.
3. The methods section (3.3. Theory and calculations) is very brief. Especially the Bayesian change point detection should be explained in more detail. Please also add a reference to the software you used for calculating the linear regression models.
We explained the Bayesian changepoint detection in more detail and added a reference to the lm-function (R Core Team, 2020) used to calculate the linear models.
4. Comparison of soil moisture measurements with modelled uFC: a) When using the (modelled) usable field capacity (uFC) provided by DWD I wonder why you did not try to make these data more comparable to the volumetric soil moisture measurements from the two lysimeters. This could either be done by converting the modelled uFC into volumetric soil moisture making use of the soil properties (in particular pF values, pore volume) these calculations are based on - as far as these information are provided along with the modelled uFC data. Or do it the other way round and calculate uFC at the soil moisture sampling points based on the volumetric soil moisture content and the soil properties (e.g. layer specific pF values) of the soil layers of the two lysimeters. This touches the above-mentioned missing information on soil properties. b) A discussion on how well modelled uFC can be compared with soil moisture measurements at the the lysimeters is missing. Presumably, the modelled uFC is based on non-artifical soils,

but soil moisture observations at the two lysimeters represent conditions in an 'artificial' soil layer. Please include a more detailed discussion here, or skip the modelled uFC data, if the soil properties on which the calculations are based are not comparable the conditions at the lysimeters.

The soil parameters used to model uFC data are different to the soil used in the lysimeter. Comparison of modeled and measured values is not possible quantitatively. However, as noted in the manuscript, they share similar temporal distribution patterns. We added information on the soil and boundary conditions used in the model to the manuscript.

5. Presentation of results: Figure 2 (discussed chapter 4) is hard to read and the information might therefore not reach the reader. Since there is many data 'squeezed' into this figure I find it hard to read or to really compare the different NP measurements. It is particularly difficult for the NP data of Field 1. Can think of another way of presenting the data, or (this goes more to the Editor) place this figure in landscape format? It might also be worth to plot it in a different way, e.g. calculate the difference from average for each depth increment over the entire observation period for each pixel/time step. I suggest to remove the map with modelled uFC at the bottom of Fig. 2 completely, or to move it to the appendix.

As suggested we moved the modeled uFC to the appendix. The Figure containing all measured soil moisture data was also moved to the Appendix and replaced by a figure showing selected soil moisture at two measurement points, along with the requested discharge (measured at lysimeter), precipitation and evapotranspiration (measured by the DWD) data. We hope the reduced amount of data helps readability.

[Figure]

L. 149: explain climatic conditions 2003 (dry and hot summer in the study region) – this can be accompanied or underlined by further general information on climate characteristics at the study site over the study period (see also #2 of my general comments above)

A figure showing annual precipitation and temperatures was added and a description of exceptional years given in section 2 "Study site" of the manuscript.

L. 161 – 165: give a more detailed description on soil properties, and discuss the effects of soil compaction. Could a compaction horizon result in a capillary barrier in the soil layer? How would that effect soil water movement?

Additional information added to section 2 of the manuscript.

Usually a capillary layer is formed by a fine material on top of a coarse material with a sharp contact. This sharp contact is not present within the compacted layer (porosity follows a gradient). The sharp contact with the layer on top is inverse to the one usually found within a capillary barrier (lower capillary forces in top layer due to higher porosity).

L. 193 – 199: please provide a more in-depth discussion in this paragraph on potential effects of soil compaction and why moisture patterns at some depths are more persistent than in other depth. E.g. continuously 'wet' conditions at approx. 100 cm in field 2 or at roughly 150 – 200 cm at field 1; why are there drier conditions in a small area above 150 cm at field 1).

added an explanation to this section regarding different compaction and soil material used during construction.

L. 203 – 206: In this paragraph you argue that the shorter observation period in field 2 is reason why the observed decrease in soil moisture is not significant. I wonder if this is the only way of interpreting this result. A) When looking at Fig. 5 there seems to be a change in the direction of soil moisture change at approx. 70 cm at field 2. This might also correspond to the rooting depth (in case vegetation is present), resulting in a quick recycling of precipitation via root water uptake / evapotranspiration which does not allow percolation to deeper soil layers. B) A compaction layer might further impede percolation. C) The different soil depths of field 1 and 2 and the different duration since the lysimeters have been installed, resulting effects on soil properties and (vegetation) cover should be discussed, too.

It certainly is not the only way of interpreting this result. There might be a correlation between rooting depth of plants with quick recycling of water in the upper soil and different trend in moisture change in deeper soil layers. However, both lysimeters have similar vegetation and thus should show similar results. It seems more likely that this effect is caused by the length of time series and differing compaction/differing material in both lysimeters.

L. 209 – 212: please provide a more detailed discussion on the reasons for the observed reductions in soil moisture (e.g. in the context of precipitation / temperature regimes). Why is the reduction in the lowest part of the soil profile most pronounced from January – May in field 2, and why is it not obvious in field 1?

We added a more detailed discussion and the precipitation data to the manuscript. Highest absolute moisture is observed during these month at the mentioned lower part in Field 2 at the beginning of the time series (largest seasonal amplitude). Maximum values of soil moisture are affected more by drying of the soil. Reduction is not obvious in Field 1 because depth of the soil in the lysimeter is less.

Figure 7, time series decomposition: this analysis is valuable to detect trend changes. As with Figure 2 I am concerned that, with the amount of data, the figure is still readable. As suggested before, it might be worthwhile to test different colour schemes, or highlight particularly relevant results.

We moved this figure to the appendix and replaced it with a figure showing only a selection of the results at two measurement points, greatly reducing the amount of data that is presented. Furthermore, to familiarize the reader with the results of the model, an example of model results is presented in an additional figure as a "more traditional" line graph and a discussion added.

L. 262: it is the first time in the manuscript that re-wetting from groundwater is mentioned. Are the lysimeters connected to the groundwater in this specific setting on a landfill? If so, please also describe this in the study site description

Lysimeters are not connected to the groundwater. Both lysimeters are constructed using plastic sealing liners at the sides and the bottom.

L. 270: which soil properties do you think of? Please include that in more detail in this paragraph

These could be any of the properties that are related to moisture transport and retention in the soil.

Added some examples for these soil properties to the manuscript: "(water retention, preferential flow paths, hydraulic conductivity, soil structure, etc.)"

Technical comments:

Chapter 1

L. 27-28: which were the effects of El Nino? Please explain the results of the cited studies (Solander, Kolusu) in more detail (1-2 sentences) or skip it

Added short explanation of main findings: "The 2015-2016 El Niño event is associated with extreme drought and groundwater storage declines in Southafrica while at the same time in east African countries south of the equator an increase in precipitation and groundwater recharge was recorded (Kolusu et al. 2019). Similarly, Solander et al. (2020) found evidence for both, increase (eastern Africa) and decrease (northern Amazon basin, the maritime regions of southeastern Asia, Indonesia, New Guinea) in soil moisture storage depending on location."

Additionally updated reference Solander, 2019 from discussion article to published version Solander 2020.

L. 30-36: you could state which measurement type is used

Added measurement type used in this study to the introduction

L. 44-46: rearrange order of sentences (e.g. start with second sentence in this paragraph

Sentences rearranged

L. 55 'with regard' instead of 'in regard' Please check and correct where necessary throughout the manuscript: - 'depth' and 'depths' - 'In depht' vs. 'in-depth' - 'at depth of' vs. 'at a depth of'

Changed to "with regard". Also checked throughout the manuscript and corrected several other instances.

Chapter 2

Include information on soil properties (at least those that are most relevant for soil moisture movement/storage), (vegetation) cover of the landfill, and climate characteristics

Added information on vegetation to the manuscript. Unfortunately not much information on soil properties and the establishment and past development of a grass cover is available.

Explain why the two lysimeters have different soil depths

Added an explanation for different soil depth in the two lysimeter fields to the manuscript. Results from the first lysimeter suggested a stronger recultivation layer better protects the mineral clay liner from drying out and thus improves long term stability of the system.

L. 78: add year '. . .being taken in December 2000' / '.. in December of that year'
added "of that year" to the manuscript

Chapter 3
Are there more neutron probe measurement points in the lysimeters (since NP numbers start with 3, and if so, why was that data not used
There are more measurement points in Field 2. As mentioned no measurements were taken at these points. Due to settling of the cover material after construction, some of the steel pipes were bent and are not usable for measurements.

L. 90: when was Field 2 constructed?
As mentioned in the section about the study site Field 2 was constructed in 2000. Added "(December 2000)" to the manuscript.

L. 122 – 127: could you please explain more clearly what you did here?
We added another figure to make this more clear.

L. 134 – 137: please describe in more detail the Bayesian change point detection and time series decomposition: how is it done and which information does it provide?
Added more information on Bayesian changepoint detection

Chapter 4
L. 141: please indicate the mentioned clay layer in Fig. 2
In accordance with a previous comment to reduce the amount of information we changed Fig. 2. It now only includes the recultivation layer.

L. 155 – 157: it is hard to see the discussed results in the Figure in its current form (see #5 in general comments)
changed figure

L. 157: change '. . .the missing occurrence. . .' into a less complicated sentence
changed

L. 160: delete 'or so'
deleted "or so"

L. 168: delete duplicate 'the bottom of'
deleted "of the bottom"

L. 170: depending on the DWD station, there is an annual cycle of uFC at 60 cm depth, so I would not call it 'minimal'
As suggested by reviewer 2, we added an indication to the boundary condition used at the bottom of the model defined as a constant water content. So annual cycle in soil moisture at the model bottom should not be very high.

L. 171: 'at a depth'
added "a", new sentence is "… is only minimal at a depth of 60 cm."

L. 172-173: Why is there a clear change in modelled uFC already after 2001, but for the soil moisture measurements this is only visible after 2003?
because we moved the time series of depth dependent usable field capacities to the appendix we removed these lines from the results and discussion section.

L. 177: add 'and a mean time series from all sampling points at field 2 are shown'
added to the manuscript according to suggestion.

L. 178 and 183: change 'in the individual time series' to 'in the time series of NP at 170

cm'

changed in both lines as suggested

L. 190: change 'could be observed' to 'soil moisture decreases by 0.34 % . . .'

Changed sentence accordingly

Figure 3: even with a good colour printer it is difficult to discern the colours representing the different years in the polar coordinate system. I suggest to try out other colour or gray scale palettes, or you just highlight very dry or very wet years.

The main reason to show data in a ploar coordinate system was to highlight the seasonal asymmetry (the opening of the nautilus representing fast re wetting). The offset from the center illustrates seasonal dynamics. Different years under dry and wet conditions can be discerned by looking at the traditional time series plot.

L. 202: change end of sentence to 'are indicated by a marker'

end of sentence changed accordingly

L. 208: I would not use 'bias' in this case, since the differences are not artefacts

bias replaced by "push" in the manuscript.

L. 215 – 218: please discuss in more detail why less water is percolating during winter in recent years

This is a direct consequence of the reduced soil moisture. We added percolation data to the manuscript.

L. 226: typo in the DWD station code?

2935 changed to correct code 3925

L. 228: 'increase' instead of 'increases'

changed

---

## Author Comment (AC2) · 16 Sep 2020

**RC2**

The manuscript presents an interesting topic and shows long-term measurements of soil moisture at a municipal landfill site in Germany. The data covers a relatively large period with quite distinct climatic conditions (wet, dry years) and measurements on soil moisture are available for several depths and various profiles at the site. Despite the rich data set I found it difficult to read, because of the uncomplete description of the experimental data and the used methods in this study. The unclear description of the lysimeter/field cover and drainage data of each filed makes it difficult to interpret the results. The authors should include at least information on the land surface cover and their change or development over time. The same should be done for the drainage data and interactions between climate, vegetation, and soil should be investigated and discussed. Hence I recommend the authors to include more data and consequently explore their data more in deep! In addition the authors should include a discussion of their results in the context of other study in the Results and Discussion section. Nevertheless it was an interesting read and I want to encourage the authors to carefully rewrite, revise and improve their manuscript.

We are thankful to Reviewer 2 for the valuable insights and the classification of the manuscript as an interesting read. We are also thankful for the more critical questions that helped to further improve the manuscript. Replies to the specific comments are given below. Description of the field site was expanded upon in the manuscript as well as additional data on precipitation and discharge added and discussed.

Specific comments:

L13: Soil moisture is not a flux. Please reformulate the sentence.

Reduced the sentence from "soil moisture fluxes" to only read "moisture fluxes".

L28: Change evaporation to evapotranspiration.

changed

L31-36: Che authors may include also the discussion about soil moisture measured by different method see e.g. Jackisch et al. (2020).

A citation to the discussion in Jackisch et al. (2020) on the different sensor systems is now added to our introduction: "A comparison and discussion of several sensor systems using different measurements principles is given in Jackisch et al. (2020), highlighting also the need for thorough calibration before the use of such systems."

L39-40: Change eg. to e.g.

changed

L49:High precision and high temporal resolved measurement with lysimeter are also able to exactly determine incoming water at the land surface due to precipitation and non-rainfall-events like dew or fog. I suggest to add this point here.

Added this important point and a corresponding reference to (Groh et al., 2018).

L53: Not clear if observations from one or two lysimeters are used in this study? Please clarify this point.

Measurements from both lysimeters were used in this study. Lysimeter Field 1 consists of one field, while Lysimeter 2 is subdivided into two separate fields that were built at the same time (see Fig. 2.). Changed this line in the manuscript to represent this fact more clearly.

L56-67: At this stage it is not clear to me why the authors need lysimeter in this study. Neither the introduction text nor the objectives are link to lysimeters. Please clarify this point! If not any soil profile with long term soil moisture measurements could be taken here instead of a more sophisticated measurement set-up with lysimeters.

We added discharge and precipitation data to make better use of the more complicated setup of the lysimeter compared to other soil moisture measurements. As the reviewer

points out, any long term soil moisture measurements could be taken here instead. We included modeled soil moisture data provided by the DWD (for a differing soil type to our measurements, thus making them not directly comparable).

Bayesian time series decomposition of further soil moisture time series from different sources (e.g. the ones published by ESA based on remote sensing) would most certainly yield interesting results, but are beyond the scope of this study.

L73-82: Please provide a table with detailed info on soil properties for all fields. This includes not only the basic info on soil texture but also other important information's, which are normally available at such municipal landfill experimental sites.

We agree that comprehensive information on used materials is important in interpreting results and added the available information to the manuscript. Unfortunately, detailed properties of the soil are not available. The major point of the monitoring program and reason for building of the lysimeters has been and still is the proper functioning of the landfill cover to stop water from percolating through the landfill itself. The material used as recultivation layer was only of minor importance during construction. Overall the soil is very heterogeneous, containing clay, sand and even larger rocks. Taking soil samples now would create cavities and change the overall behavior of the lysimeter, which is undesirable.

L85-86: The authors should explain the modification of the measurement devices in detail, if not study results might not be comparable to other studies.

The diameter of the probe was reduced to fit within the installed steel pipes. This does not influence the measurements. Both neutron probes were calibrated (Augenstein, 2015).

L93: How does this number fit with the mean inclination angle of 23.5 ∘ for each field reported in L75 & L79?

Final depth is not dependent on inclination angle. Exchanged "bottom" by "final depth" to alleviate the confusion. Depths given indicate distance from surface to bottom of the recultivation layer. The inclusion of a figure showing lysimeter cross sections should make this more clear.

L102: The authors should explain how the used uFC data were derived for this study. This includes the important assumption e.g. vegetation, model, boundary conditions and soil types/ properties.

The data are provided by the DWD as is. Added an explanation on their calculation and boundary conditions to the manuscript.

L102-107: Totally unclear why the authors want to use the model uFC? This should be explained in the section.

As noted by the reviewer in a previous comment, any soil moisture time series could be used. We used external data by the DWD to compare with measured soil moisture and validate our findings. Added this explanation to the manuscript.

L103: Which soil types are used for this uFC data? The authors should describe the soil properties to be able to compare it with the landfilled soil profiles.

Added a description.

L109-116: The authors should clarify why time series were transformed into a radial coordinate system. This was done only for soil moisture observations?

We added the reason given in the results section to the calculations section. Yes.

L117: Explain more in detail why the authors used linear regression and for what. Did the authors also check if the assumptions for using such a model are full filled?

Before experimenting with more sophisticated models that are often unnecessarily complex and hard to interpret, we tried to gain insights by using well known methods. Linear regressions are widely used and they are easy to formulate and calculate.

Decline in soil moisture does not follow a strictly linear trend over the complete observation period (as mentioned several times throughout the manuscript). Furthermore, a large seasonal component is superimposed on the overall trend. We still think our use of linear regressions is useful to gain insights and that our careful attempt at interpreting results (overall decline in soil moisture) is valid.

L121: What happens in leap years? Why using 365.2425 instead of just using the length of the corresponding year, which can be 365 or 366 days long!

No measurements were taken on 31. Dez. (day 366) during leap years, so no overlap between years occurs. The difference between the two calculations is only minor and does not affect overall interpretation. For calculation into the radial coordinate system, we changed calculation to be based on lengths of individual years.

As for the slope of the regression lines, these span over multiple years and thus an average length for a year was used.

L134-137: Please explain the used method i.e. "Bayesian change point detection" and "time series decomposition" used in this investigation more in detail! The used methods should also be included in the introduction section and it should be shown why this kind of methods are appropriate for such an investigation.

We added a short justification for the use of the Bayesian change point detection model and an explanation for the underlying methodology. A more detailed description is presented in the cited literature and does explain the intricacies of the model more thoroughly than we can present them in our manuscript.

L139: Please be precise: Figure 2 shows the monthly soil moisture profiles at the corresponding position, which were derived from the single measurements. I recommend also to add a) to the soil moisture and b) to the uFC subplot. I suggest in addition to use the same coloring scheme for both subplots. This makes it easier for the reader to compare between soil moisture and uFC.

The plot shows all measured soil moisture data before monthly averages were calculated. We changed this sentence to be more precise. Furthermore, we moved the complete figure to the appendix and replaced it with a new one showing only a selection of the soil moisture data plus some additional precipitation and discharge data in an effort to increase comprehension.

L142: Please explain why only RL will be evaluated?

We did this because it reflects best the processes and moisture dynamics found in natural soils. The mineral clay liner, capillary barrier system and functioning of the sealing system as such are not representative of processes in natural soils.

L152: The authors should show this recharge data for each lysimeter in a separate figure! In addition to that the authors might show the precipitation data in the figure. Please discuss the different conditions during the observation period e.g. dry years, wet years and its implications for the observed soil moisture.

We added precipitation data provided by the DWD as well as measured discharge from both lysimeters. We further added information on especially wet and dry years.

[Figure]

L153: Re-wetting 2018. This might be related to the in general wetter conditions in 2017! The authors should explore their data more in depth!

It is true that precipitation in 2017 was above average as a whole, and especially during the second half of the year. At the time the year 2017 did not stand out as a particularly dry year and therefor the very low soil moisture in lower soil layers was surprising. Precipitation at the change of the years 2017/2018 percolated through the soil column in the lysimeter leading to measurable discharge re-wetting of the soil at the beginning of 2018.

L139-165: The general patterns of the soil moisture can be seen relatively well from the figure 2. However, other results discussed here are difficult to see from this figure. I suggest the authors to re-think what the main purpose for showing the figure here is and change it in a way that main findings are clearly visible.

We replaced the figure by one that is hopefully easier to read, and also contains additional data on precipitation and discharge, but reducing the number of soil moisture data shown.

L139-165: Please discuss results in the light of soil properties at each plot and the vegetation of these fields/lysimeters.

Both lysimeters share same soil and vegetation cover.

L168: Please report which model bottom boundary was by the DWD to simulate uFC and discuss this in the light of the presented uFC values.

The model used by the DWD uses constant water content as boundary condition at the bottom of the model. At the upper boundary, precipitation and evapotranspiration are used to calculate water content.

L171: Not sure from which observations I can see evaporation depths over 200cm from Figure 2? Please explain your findings more in detail! Augenstein et al. (2015) reports that the fields are covered by grass, so the authors should discuss also here the vegetation development of the lysimeters/fields and refer in the manuscript consequently to evaporation and transpiration. Was there any change in the vegetation over the observation period? From my perspective higher soil moisture values at the beginning of the period might be rather related to the establishment of vegetation on the fields i.e. change from bare soil with only evaporation to a field cover with grass including evaporation and transpiration. Please clarify this point!

Regarding your first comment raised on the visibility of evaporation depth. This can be seen from the measurements at the lysimeter and the seasonal pattern visible at these depth, not the modeled data.

Regarding your second comment on the vegetation cover. Both lysimeters are covered by grass. Further information on the development of the vegetation cover is not available. Nonetheless, the reviewer raises an interesting point here on the evapotranspiration being higher under dense vegetation cover compared to bare soil. This could indeed be the case here or at least a contributing factor. It has to be noted, however, that even the modeled data show this change in behavior at around the same time. If indeed the change in soil moisture is a result of a change in vegetation, this change must (at least in part) be driven by the factors included in the model, mainly meteorological parameters (precipitation, temperature, radiation). Changes in measured soil moisture at around the year 2003 could also be the result of the establishment of a vegetation cover after the construction of the lysimeter and over several consecutive years. The soil cover is important to prevention of erosion and lowering overall percolation by increasing evapotranspiration. The system is designed with a vegetation cover as an integral part to it's proper functioning. Furthermore, Field 1, which has been constructed several years prior to Field 2 shows a similar change. And, as mentioned, a similar change is visible in the modeled data. It is still possible that vegetation and evapotranspiration both drive this change, but then it has to be connected through the meteorologic parameters used in the model (e.g. longer vegetation periods).

L171: After looking at Augenstein et al. (2015) the authors should also clarify in the M&M section that the depths across the inclined field varied. In addition the authors should include the info that layers of the profiles e.g. in field 2B are not the same missing mineral clay liner (referring Fig. 1b in Augenstein et al. (2015))

This is true. We also added cross sections of the lysimeters to make this more clear.

L176: For a better comparison of the time series I recommend to put both in on plot. This makes it easier to compare.

The time series serves also as a reference for color in the radial plots. Having both in one plot would require different color scales for the two time series. To conserve this reference to color and at the same time improve comparability between the two, we now show both time series in both plots by adding the respective other in grey.

[Figure]

L177: Mean soil moisture of what depths or measurement profiles?

Mean soil moisture of the recultivation layer in Field 2 was calculated as average of NP3 at depth between 10 cm and 180 cm and NP5, NP6 and NP7 at depth between 10 cm and 220 cm. This is the same depth range that is used and presented in Fig. 5 and Fig. 7.

L175-181: It would be interesting to see this time series for the field 1 as the time series of this plot is much larger than for field 2.

The mentioned asymmetry is much more pronounced in time series from Field 2. As noted in section 3.1., measurements for Field 1 were only taken monthly after Field 2 was built. This reduced temporal resolution might be one factor affecting this. Additionally, the recultivation layer in Field 2 is much thicker than in Field 1 and downward propagation of the moisture front is spread out over a longer time frame. The time series for Field 1 is given below.

[Figure]

L175-181: First: I can see from the time series specific changes after the extremely dry year 2003. Please discuss here possible reasons! You might have a look at e.g. Robinson et al. (2016) or Groh et al. (2020), which showed within their investigations a change in the soil moisture level after drought events. Rahmati et al. (2020) showed for two grassland site a trend of decreasing seasonal minimal soil moisture after drought event in 2015. I guess there is much more literature on that point and I suggest the authors to include a more profound discussion/comparison of their findings.

We agree that discussion of the changes in soil moisture following the extremely dry year 2003 is important and want to thank the reviewer for the suggested literature that was a great help in this. We expanded on our interpretations given in the section on time series decomposition (4.4) and included some additional references (Robinson et al. 2016, D'Orico et al. 2007).

L139-187: I recommend the authors to use additional methods to analyze the soil moisture time series. It would be worth also to include time series of precipitation and potential evapotranspiration. The authors could also look at the relations between those variables and soil moisture observations e.g. by Wavelet-analysis (see e.g. Graf et al., 2014; Bravo et al., 2020; Rahmati et al., 2020).

We added information on precipitation and evapotranspiration to the manuscript. Relations between variables might indeed be very interesting to look at in a future study.

L-Figure 2: The authors should explain visible artefacts, i.e. strange lines between 2007 and 2008, white points, and strange lines in 2004 [. . ..].

White points are missing data. Due to external circumstances it can sometimes happen that measurements in the field have to be aborted, leaving gaps in the data.

Strange lines are most likely artifacts caused by faulty data entry into the data base. It can also be seen in Augenstein et al. (2015) and was left as is.

L188 & 200: Did the authors check the important assumptions associated with a linear regression model? The questions arises as I can see a change in soil moisture level after drought event in 2003, which might affect the distribution of the data.

This change in soil moisture is mentioned many times in the manuscript as well as the influence that the length of time series has as a result. We are aware of the assumptions made in linear regressions and considered these during interpretation.

L198-199: Please reformulate the sentence.

Sentence reformulated

L201: Please show also the values for field 1 below 100 cm in figure 5.

Only results for the recultivation layer are shown. The recultivation layer in Field one only has a thickness of 100 cm.

L205-206: Please explain in detail why data for field 1 before 2003 where excluded here.

All data are included in the analysis discussed in this passage.

L207-208: Not sure why the inclusion of data before 2003 would bias the results of field 1?

Because soil moisture was significantly higher at the beginning of the measurement series. The graph below (also shown in Appendix of the discussion article) shows the influence that the length of the time series used has on resulting soil moisture change based on linear regression. Exclusion of data pre 2003 leads to reduction in R² values.

[Figure]

L215: The authors should show and discuss this percolation data.

Data added to the manuscript (see above)

L215-218: The authors should clarify to which field this results are related.

Results are related to both lysimeter fields. Added this information to the manuscript.

L219-229: The authors should as already mentioned provide the background info of this model simulation in the M&M section. This is important to better understand and especially discuss the results. So e.g. which vegetation was used in the simulation, which model, does this model provide a coupling of plant and soil dynamically or use of a fixed LAI and so on.

Added information on the model used by the DWD to the manuscript

L227: Unclear how the authors come to this conclusion. Please clarify this!

We opted to exclude this conclusion.

L230: I could not fully evaluate this section as there is very few information on the used methods in the M&M section.

Added more information on this method to the manuscript in the corresponding section

L235: The authors should explain why 2003 was that important for the soil moisture and actually discuss reason for this observations. Please do this in the whole manuscript.

2003 was exceptionally dry leading to lower soil moisture levels in the following years. Feedback mechanisms between soil moisture, temperature, precipitation and evapotranspiration are probably some of the reasons.

L238: Is this related to climatic conditions, evolution of the land surface cover or due to changes in the soil after packing the lysimeter? Please clarify this point! The authors also should be aware that landfill soils might behave different than natural developed soils.

This could be related to both, external factors like climate and internal factors regarding the lysimeter. But the temporal changes of soil properties in the lysimeter and vegetation cover have not been recorded. So we currently don't know.

L262: Yes indeed that might be a reason! This is actually the first line where discussion of the results starts! However I want to point out that current lysimeters might overcome such issues as those systems are able mimic not only a more dynamical recharge but also the capillary rise from shallow groundwater or deeper soil layers. For further details on this lysimeters see Unold and Fank (2008); Pütz et al. (2016); Herbrich et al. (2017); Groh et al. (2020); and the effect of shallow groundwater table on land surface water fluxes Kollet and Maxwell (2008); Groh et al. (2016).

The lysimeter was built in 1993 and does not provide this capability.

L263:

This is not truth as the model used by the DWD accounts also for capillary rise. See https://www.dwd.de/DE/fachnutzer/landwirtschaft/dokumentationen/allgemein/ bf_erlaeuterungen.pdf;jsessionid=C44 at the chapter "Hintergründe zum Modell".

The DWD model assumes constant water content at the bottom boundary. Discussion was changed accordingly.

L271: I could not find any data in the manuscript that actual shows that hysteresis plays are role at this site. So please clarify the following sentence: "There are clearly hysteresis effects during drying and re-wetting of the soil".

Section 4.1 is dedicated to the asymmetry of drying and re-wetting. This asymmetry could also be called a hysteresis, as drying and re-wetting do not follow the same temporal paths.

Additionally, we added a citation to Augenstein et al. (2015). They investigated and found evidence for the hysteresis between soil moisture and discharge.

L278-280: Very vague statement! Please discuss this in a broader context and compare findings with other studies!

Expanded discussion and added references to further studies.

L300: Not sure if the observation provide the info if this processes are irreversible or reversible! Please discuss this before in the Results & Discussion section.

By use of the word "permanently" we did not necessarily mean "irreversible". We reformulated the sentence to avoid this word and leaving the question of reversibility open. Although other studies described similar phenomena as irreversible.

L301: I am confused about this statement as the authors used a simple linear model in this manuscript!

While it is true that we used a lot of simple linear models, ( that are unable to detect changes in the overall dynamics of trend and especially seasonality) we also applied a Bayesian model to detect changepoints in trend and seasonality during time series decomposition.

L301-303: That's truth! Thus I recommend the authors to include also vegetation and drainage data to further explore their already rich data set and to include possible interactions between land surface cover, soil moisture and drainage.

We added available data on vegetation and drainage to the manuscript.

Augenstein, M., Goeppert, N., Goldscheider, N., 2015. Characterizing soil water dynamics on steep hillslopes from long-term lysimeter data. Journal of Hydrology, 529: 795-804, https://doi.org/10.1016/j.jhydrol.2015.08.053. Bravo, S., González-Chang, M., Dec, D., Valle, S., Wendroth, O., Zúñiga, F., Dörner, J., 2020. Using wavelet analyses to identify temporal coherence in soil physical properties in a volcanic ash-derived soil. Agricultural and Forest Meteorology, 285-286: 107909, https://doi.org/10.1016/j.agrformet.2020.107909. Graf, A., Bogena, H.R., Drüe, C., Hardelauf, H., Pütz, T., Heinemann, G., Vereecken, H., 2014. Spatiotemporal relations between water budget components and soil water content in a forested tributary catchment. Water Resources Research, 50(6): 4837-4857, 10.1002/2013WR014516. Groh, J., Vanderborght, J., Pütz, T., Vereecken, H., 2016. How to Control the Lysimeter Bottom Boundary to Investigate the Effect of Climate Change on Soil Processes? Vadose Zone Journal, 15(7): 1-25, 10.2136/vzj2015.08.0113. Groh, J., Vanderborght, J., Pütz, T., Vogel, H.J., Gründling, R., Rupp, H., Rahmati, M., Sommer, M., Vereecken, H., Gerke, H.H., 2020. Responses of soil water storage and crop water use efficiency to changing climatic conditions: a lysimeter-based space-for-time approach. Hydrol. Earth Syst. Sci., 24(3): 1211-1225, 10.5194/hess-24-1211-2020. Herbrich, M., Gerke, H.H., Bens, O., Sommer, M., 2017. Water balance and leaching of dissolved organic and inorganic carbon of eroded Luvisols using high precision weighing lysimeters. Soil and Tillage Research, 165: 144-160, 10.1016/j.still.2016.08.003. Jackisch, C., Germer, K., Graeff, T., Andrä, I., Schulz, K., Schiedung, M., Haller-Jans, J., Schneider, J., Jaquemotte, J., Helmer, P., Lotz, L., Bauer, A., Hahn, I., Šanda, M., Kumpan, M., Dorner, J., de Rooij, G., Wessel-Bothe, S., Kottmann, L., Schittenhelm, S., Durner, W., 2020. Soil moisture and matric potential – an open field comparison of sensor systems. Earth Syst. Sci. Data, 12(1): 683-697, 10.5194/essd-12-683-2020. Kollet, S.J., Maxwell, R.M., 2008. Capturing the influence of groundwater dynamics on land surface processes using an integrated, distributed watershed model. Water Resources Research, 44(2): W02402, 10.1029/2007WR006004. Pütz, T., Kiese, R., Wollschläger, U., Groh, J., Rupp, H., Zacharias, S., Priesack, E., Gerke, H.H., Gasche, R., Bens, O., Borg, E., Baessler, C., Kaiser, K., Herbrich, M., Munch, J.-C., Sommer, M., Vogel, H.-J., Vanderborght, J., Vereecken, H., 2016. TERENO-SOILCan: a lysimeter-network in Germany observing soil processes and plant diversity influenced by climate change. Environmental Earth Sciences, 75(18): 1-14, 10.1007/s12665-016-6031-5. Rahmati, M., Groh, J., Graf, A., Pütz, T., Vanderborght, J., Vereecken, H., 2020. On the impact of increasing drought on the relationship between soil water content and evapotranspiration of a grassland. Vadose Zone Journal, 19(1): e20029, C810.1002/vzj2.20029. Robinson, D.A., Jones, S.B., Lebron, I., Reinsch, S., Domínguez, M.T., Smith, A.R., Jones, D.L., Marshall, M.R., Emmett, B.A., 2016. Experimental evidence for drought induced alternative stable states of soil moisture. Scientific Reports, 6: 20018, 10.1038/srep20018. Unold, G., Fank, J., 2008. Modular Design of Field Lysimeters for Specific Application Needs. Water Air Soil Pollut: Focus, 8(2): 233-242, 10.1007/s11267-007-9172-4.

---

## Referee Report (RR1)

The authors have revised the paper according to the comments of the first review. Below are the remarks that I have on this revised version. In general, I agree that the long time series of soil moisture measurements is valuable and worth publishing. However, my concern remains that a) the interpretation and in-depth analysis of the data is restricted by the fact that information on soil properties is lacking and b) the study represents a very specific case and the interpretation of results is limited to these specific conditions at the landfill. I suggest to underline this in the manuscript.

Fig. 6: mark individual figures from a) to g). Is it monthly precipitation? Pleas add this info in the figure caption. Same with the blue line (mean monthly precipitation?) Explain the discussion of Fig. 6 why uFC is partially >100%?

L185. "...because it is thought to reflect best the processes and moisture dynamics found in natural soils". From the site description you give, it seems that it's not close to a 'natural soil'

L. 189ff: what is meant by discharge here? Discharge measured at the bottom of the soil profile, or movement of water through the soil profile?

L. 192ff: NP5 instead of NP3?

L.201ff: which Figure are you referring to? NP9, NP12 are not shown in Fig. 6; same with MCL of Field 1

L. 209f: "Porosity and hydraulic conductivity is therefore not uniformly distributed over the complete depth of the lysimeter". I think that holds true for most soil profiles. The consistent and very distinct break of soil moisture over the entire measurement period rather suggests that there is a distinct change in porosity and hydraulic conductivity between the two layers (i.e. lower porosity at the top of the lower soil layer

L. 212f: "Settling down of the soil cover in the years after construction may additionally change soil properties over time." The soil moisture break remains consistent over the measurement period. Soil properties may have changes over time, but this is not reflected in the data you present. What about the consistently lower soil moisture values between approx. 100 – 150 cm, and the higher soil moisture at the bottom of the RL? Please discuss this also in the text.

Fig. 7: explain the different lines in upper and lower graph (gree-blue line and grey line). What is the data unit at the polar coordinate graph? [%] (as from Eq. 1 / 2)? What do the negative values stand for?

L. 221-225: This new section is more appropriate in the Discussion section. Last sentence is not clear, please rewrite

L. 233: Is there a time lag in the measurements or a lag in the propagation of soil moisture in the profile? Please restructure/clarify this sentence.

L. 251: coefficients

L. 260: add "Resulting slopes with p > 0.05 (i.e. soil moisture change is not significant) are indicated…"

Fig. 9: add info that upper graph is for Field 2 and lower for Field 1 in caption

L. 281: months

L. 312: Sentence can be skipped because it was mentioned before: "Measurements at Field 2 (NP 3, NP5, NP6, NP7) have started later compared to Field 1."

L. 377: when  applied

---

## Author Response (AR2)

**Response letter concerning the manuscript hess-2020-274.**

**hess-2020-274-referee-report-1**

Dear Markus Merk and co-authors,

the manuscript has partially improved, but as already mentioned in the first revision of the draft, there is nearly no discussion, broader interpretation and comparison to other studies of the results in the section 4. In addition, drainage data that is now shown are poorly evaluate in the context of the main message of the manuscript. I think thes data can nicely show/help to quantify the impact of SWC depletion on recharge from this study. Many other study can not do this, because the don't have the data from lysimeter. So I want to encourage the authors to add this analysis in a sub-section of Results and Discussion section, which would be an important information for the water sector, especially in the context of climate change.

I recommend again that the manuscript section 4 needs a broader discussion, interpretation and comparison of the results before publication in HESS.

We would like to express out thanks to the reviewer for valuable suggestions! We would be happy to share the data for further use in additional studies after publication. However, further analysis of the drainage data and discharge behavior is beyond the scope of this study.

General

1) It was difficult to follow up the response in text on the review, as authors did not adapt lines in the response from the revised version of the manuscript.

We would like to thank the reviewer for his time and effort to diligently consider our responses.

2) I recommend adding a section where the authors link the soil moisture depletion with the drainage data. Many other studies lack on the this information and the authors should use the available data from the lysimeter to link impacts of changing soil moisture on drainage/ recharge which is of high importance for the water sector

We agree that this information is valuable and even more could be learned from our data. However, further analysis of the drainage data and discharge behavior is beyond the scope of this study.

Specific comments:

L99: Missing units in the following sentences: "For the year 2002, the porosity of the RL is 0.4, usable field capacity 0.25 and the wilting point at 0.14. The 100 permeability was estimated as kf = 1:0 10 6. Values were observed to be variable over time."

added the appropriate units: porosity [-], usable field capacity [-], wilting point [-], permeability [ms$^{-1}$]. (L.99-100)

L100: Please explain the statement that values were observed to be variable over time.

The determined parameters were used to model discharge from the lysimeter using weather data. In subsequent years, these parameters were changed (taking the range of determined uFC, wilting point, etc. into account) to calibrate the model. We changed the statement to read "Formation of preferential flow paths in the lysimeter lead to changes in hydraulic properties over time (Gerlach, 2007)." (L.100-101)

L128: Again, please provide units for the given wilting point and field capacity

Added appropriate units: [-]. (L.129-130)

First response Review round 1 L102-107: Totally unclear why the authors want to use the model uFC? This should be explained in the section.

   As noted by the reviewer in a previous comment, any soil moisture time series could be used. We used external data by the DWD to compare with measured soil moisture and validate our findings. Added this explanation to the manuscript.

     Response: Validating measurements with external simulation data?

The changepoint in the year 2003 has been observed before, but it was thought to be the result of an initial draining of the lysimeter after construction. The discovery of similar behavior in modeled soil moisture (e.g. uFC) led us to investigate this further. Some groundwater level time series also indicate a change that happened in 2003 with a subsequent decline groundwater levels.

L171: Change Zhao et al. 2019b to Zhao et al. 2019a, and change vice versa in line 176 Zhao et al. 2019a to Zhao et al. 2019b

Quotations are in accordance with the manuscript preparation guidelines.l

L240: Figure caption of Fig. 6. "real evapotranspiration". Please use in this context actual evapotranspiration, because there is "no unreal" evapotranspiration. It is also not clear to me where this data is coming from and if it is an outcome of the simulation from the AMBAV model (than please refer to simulated actual evapotranspiration) or if it is indeed measured actual evapotranspiration EC-tower, lysimeter? Please clarify this please

The data are from the cited source. The term "real evapotranspiration" is used in the official documentation by the DWD:

ftp://opendata.dwd.de/climate_environment/CDC/derived_germany/soil/daily/historical/

We changed the figure caption to read "Simulated monthly averages of usable field capacity (loamy sand), simulated potential evapotranspiration (red) and simulated actual evapotranspiration (black) (DWD Climate Data Center, 2020)..." (Fig. 6 caption)

L199: reformulate this sentence, as low precipitation does not lead to a drying out. It is the large atmospheric demand for ET paired with low P [....].

Sentence reformulated "...below average, and paired with a large atmospheric demand for ET, once again drying out the lower soil..." (L.201)

L201: not clear from which figure I can actual see this?

Added reference to Fig. A1. (L.204)

L180-227: I could not find any discussion of their results in this section, also newly added discharge in the Figure is not presented in the text nor its effect from depleting soil water storage/ decline in soil moisture profile on DL is given. In addition, the potential and actual evapotranspiration is not present in the text and discussed.

Further analysis is beyond the scope of this study.

L242: thanks to add the other SWC time series in the corresponding subplot (gray lines), but the authors should mention this in the legend of the figure and also in the text figure caption.

We added a legend to this figure and amended the figure caption accordingly. (Fig.7 and Fig.7 caption)

First response Review round 1: L271: I could not find any data in the manuscript that actual shows that hysteresis plays are role at this site. So please clarify the following sentence: "There are clearly hysteresis effects during drying and re-wetting of the soil".

   Section 4.1 is dedicated to the asymmetry of drying and re-wetting. This asymmetry could also be called a hysteresis, as drying and re-wetting do not follow the same temporal paths. Additionally, we

added a citation to Augenstein et al. (2015). They investigated and found evidence for the hysteresis between soil moisture and discharge.

Response: there is no hint on hysteresis in this section. Observed changes in SWC might be related to hysteresis, change in soil surface cover (large ET, which reduce SWC), or actually to a change of soil properties itself. Dear authors there is no discussion of the presented results.

Changed sentence to make the citation even more clear: "Augenstein et al. (2015) found, that there are hysteresis effects during drying and re-wetting of the soil at this site."
A description of this hysteresis is given in section 4.5. of their manuscript and the hysteresis between discharge and soil moisture shown in their Fig. 7. (L.331-332)

L274: the authors should have a closer look at this data, and should show recharge data from both lysimeters in a better way to connect depleted soil moisture to changes in the recharge behavior of the soils over time.
This has been done in depth by Augenstein et al. (2015) and is beyond the scope of this study.

L276: this statement is not clear to me? Did the authors mean characteristic length of evaporation i.e. this means the depth until which a wet soil dries out during stage 1 evaporation? But it's an vegetated soil so E and T occurs (see next commentary). I suggest rather that it is an impact of many different factors, e.g. high ET that reduce soil moisture and also seasonal low P, but also to other factors like changes in the vegetation cover. As this is a main outcome the results should be discussed in a much broader context and should be compared with other studies.
It is the depth at which a clear seasonal pattern is visible. We changed it to evapotranspiration as per the next comment. (L.272)

L276: Also here it is rather related to ET and not only to E.
Changed to evapotranspiration. (L.272)

L336-354: this is actually the first and only part of the Results and Discussion section, where the authors discuss their results in the light of previous studies. This need to be done in a proper way in each of the previous section, if not the autos should change in L180 Results and Discussion to Results and add after this a separate Discussion section.
Separated Results and Discussion as suggested. (L.181, L.328)

L367-368: so this indicates that changes in the vegetation cover might be the large driver of the observed depletion of SWC
As suggested we added this to the manuscript: "This indicates that changes in the vegetation cover might be the large driver of the observed depletion of soil water." (L.369)

**hess-2020-274-referee-report-2**

The authors have revised the paper according to the comments of the first review. Below are the remarks that I have on this revised version. In general, I agree that the long time series of soil moisture measurements is valuable and worth publishing. However, my concern remains that a) the interpretation and in-depth analysis of the data is restricted by the fact that information on soil properties is lacking and b) the study represents a very specific case and the interpretation of results is limited to these specific conditions at the landfill. I suggest to underline this in the manuscript.

a) We agree. But further sampling to increase knowledge of soil properties is not possible.
b) We added "The study represents a very specific case and the interpretation of results is limited to these specific conditions at the landfill." at the beginning of Results and Discussion section. (L.182)

Fig. 6: mark individual figures from a) to g). Is it monthly precipitation? Pleas add this info in the figure caption. Same with the blue line (mean monthly precipitation?) Explain the discussion of Fig. 6 why uFC is partially >100%?

We added a) to g) to individual subplots. Added information of mean monthly precipitation to the figure caption. Added a legend to subplot g). (Fig.6 and Fig.6 caption)

L185. "...because it is thought to reflect best the processes and moisture dynamics found in natural soils". From the site description you give, it seems that it's not close to a 'natural soil'

It is of course not a natural soil but an artificially built system. However, that does not necessarily mean that it behaves in a way that is different to natural soils. These can have very diverse properties. What is originally meant by this statement is, that the soil properties of the drainage layer and MCL are more different from natural soils compared to the RL and natural soils. So, out of all layers in the lysimeter, the RL is a best representation of natural soil. Sentence was changed to: "...*because it is thought to be the layer in the lysimeter that reflects best the processes and moisture dynamics found in natural soils.*" (L.188)

L. 189ff: what is meant by discharge here? Discharge measured at the bottom of the soil profile, or movement of water through the soil profile?

We added: " *and is measured as discharge from the DL.*". (L.192)

L. 192ff: NP5 instead of NP3?

We changed NP3 to NP5 because NP5 data is shown in Fig. 6. (L.195)

L.201ff: which Figure are you referring to? NP9, NP12 are not shown in Fig. 6; same with MCL of Field 1

This refers to supplemental Fig. A1. We added the appropriate reference. (L.204)

L. 209f: "Porosity and hydraulic conductivity is therefore not uniformly distributed over the complete depth of the lysimeter". I think that holds true for most soil profiles. The consistent and very distinct break of soil moisture over the entire measurement period rather suggests that there is a distinct change in porosity and hydraulic conductivity between the two layers (i.e. lower porosity at the top of the lower soil layer

We added: "*The consistent and very distinct break of soil moisture over the entire measurement period suggests that there is a distinct change in porosity and hydraulic conductivity between these two layers.*" to the manuscript. (L.213-214)

L. 212f: "Settling down of the soil cover in the years after construction may additionally change soil properties over time." The soil moisture break remains consistent over the measurement period. Soil properties may have changes over time, but this is not reflected in the data you present. What about the consistently lower soil moisture values between approx. 100 – 150 cm, and the higher soil

moisture at the bottom of the RL? Please discuss this also in the text.
See previous comment.

Fig. 7: explain the different lines in upper and lower graph (gree-blue line and grey line). What is the data unit at the polar coordinate graph? [%] (as from Eq. 1 / 2)? What do the negative values stand for?
We added a legend to upper and lower graphs. The gray line was added as visual reference by reviewer request. The data unit (%) was added to the polar graphs and the sign of negative values removed. The bottom "axis" in the polar graphs is more like a scale-bar than an actual axis. Although it is uncommon to move this axis in a way that it no longer intersects the origin point, we think in this case it is less obstructive to the shown data. (Fig.7)

L. 221-225: This new section is more appropriate in the Discussion section. Last sentence is not clear, please rewrite
Paragraph moved to the Discussion section. And last sentence changed for clarity. (L.217-223, L.341-348)

L. 233: Is there a time lag in the measurements or a lag in the propagation of soil moisture in the profile? Please restructure/clarify this sentence.
It is indeed the latter. We changed the sentence accordingly. (L.229-230)

L. 251: coefficients
Deleted the "s". (L.247)

L. 260: add "Resulting slopes with p > 0.05 (i.e. soil moisture change is not significant) are indicated..."
Added according to suggestion. (L255-256)

Fig. 9: add info that upper graph is for Field 2 and lower for Field 1 in caption
Added "Upper graphs: Field 2, lower graphs: Field 1. " to the figure caption. (Fig.9 caption)

L. 281: months
changed to months. (L.276)

L. 312: Sentence can be skipped because it was mentioned before: "Measurements at Field 2 (NP 3, NP5, NP6, NP7) have started later compared to Field 1."
Sentence skipped and changed to: Measurements at Field 2 (NP 3, NP5, NP6, NP7) have started later compared to Field 1. They also show higher initial soil moisture contents. (L.304)

L. 377: when a applied
a deleted. (L.378)

---

## Author Response (AR3)

Dear Editor,

on behalf of all co-authors I would like to thank you for this opportunity to improve the manuscript and your suggestion to further explore the drainage data. I have added a section concerning the discharge measurements to the Methods section, added a sub-section dedicated to the drainage data in the Results section and expanded the Discussion.

Best regards